# Structural and functional evaluation of *de novo*-designed, two-component nanoparticle carriers for HIV Env trimer immunogens

**Aleksandar Antanasijevic**[1,2], **George Ueda**[3], **Philip J. M. Brouwer**[4], **Jeffrey Copps**[1,2], **Deli Huang**[5], **Joel D. Allen**[6], **Christopher A. Cottrell**[1,2], **Anila Yasmeen**[7], **Leigh M. Sewall**[1], **Ilja Bontjer**[4], **Thomas J. Ketas**[7], **Hannah L. Turner**[1,2], **Zachary T. Berndsen**[1,2], **David C. Montefiori**[8], **Per Johan Klasse**[7], **Max Crispin**[6], **David Nemazee**[5], **John P. Moore**[7], **Rogier W. Sanders**[4], **Neil P. King**[3], **David Baker**[3,9], **Andrew B. Ward**[1,2]*

**1** Department of Integrative, Structural and Computational Biology, Scripps Research, La Jolla, California, United States of America, **2** International AIDS Vaccine Initiative Neutralizing Antibody Center, the Collaboration for AIDS Vaccine Discovery (CAVD) and Scripps Consortium for HIV/AIDS Vaccine Development (CHAVD), Scripps Research, La Jolla, California, United States of America, **3** Institute for Protein Design, Department of Biochemistry, University of Washington, Seattle, Washington, United States of America, **4** Academic Medical Center (AMC), University of Amsterdam, Amsterdam, Netherlands, **5** Department of Immunology and Microbiology, Scripps Research, La Jolla, California, United States of America, **6** School of Biological Sciences, University of Southampton, Southampton, United Kingdom, **7** Weill Cornell Medicine, Cornell University, New York, New York, United States of America, **8** Department of Surgery, Duke University Medical Center, Durham, North Carolina, United States of America, **9** Howard Hughes Medical Institute, Chevy Chase, Maryland, United States of America

* andrew@scripps.edu

**Data Availability Statement:** All electron microscopy density maps are available from the Electron Microscopy Databank (accession

## Abstract

Two-component, self-assembling nanoparticles represent a versatile platform for multivalent presentation of viral antigens. Computational design of protein nanoparticles with differing sizes and geometries enables combination with antigens of choice to test novel multimerization concepts in immunization strategies where the goal is to improve the induction and maturation of neutralizing antibody lineages. Here, we describe detailed antigenic, structural, and functional characterization of computationally designed tetrahedral, octahedral, and icosahedral nanoparticle immunogens displaying trimeric HIV envelope glycoprotein (Env) ectodomains. Env trimers, based on subtype A (BG505) or consensus group M (ConM) sequences and engineered with SOSIP stabilizing mutations, were fused to an underlying trimeric building block of each nanoparticle. Initial screening yielded one icosahedral and two tetrahedral nanoparticle candidates, capable of presenting twenty or four copies of the Env trimer. A number of analyses, including detailed structural characterization by cryo-EM, demonstrated that the nanoparticle immunogens possessed the intended structural and antigenic properties. When the immunogenicity of ConM-SOSIP trimers presented on a two-component tetrahedral nanoparticle or as soluble proteins were compared in rabbits, the two immunogens elicited similar serum antibody binding titers against the trimer component. Neutralizing antibody titers were slightly elevated in the animals given the nanoparticle immunogen and were initially more focused to the trimer apex. Altogether, our findings indicate that tetrahedral nanoparticles can be successfully applied for presentation of

numbers 21175–21186); associated models are
deposited within the Protein Databank (accession
numbers 6VFL and 6VFK).

**Funding:** This work was supported by grants from
the National Institute of Allergy and Infectious
Diseases Center for HIV/AIDS Vaccine
Immunology and Immunogen Discovery
UM1AI100663 (M.C., A.B.W.), Center for HIV/AIDS
Vaccine Development UM1AI144462 (M.C., A.B.
W.), P01 AI110657 (J.P.M., R.W.S., and A.B.W.),
R01 AI36082 (J.P.M.), R01AI073148 (D.N.) and
R01AI128836 (D.N.); and by the Bill and Melinda
Gates Foundation and the Collaboration for AIDS
Vaccine Discovery (CAVD) OPP1156262 (N.P.K.,
D.B.), OPP1120319 (N.P.K., D.B.), OPP1115782
(A.B.W.), OPP1111923 (D.B., N.P.K., J.P.M., and
R.W.S.), OPP1132237 (D.B., N.P.K., J.P.M., and R.
W.S.), OPP1146996 (D.C.M.) and OPP1196345 (A.
B.W.); and by the National Science Foundation
grant DMREF 1629214 (N.P.K., D.B.); and by the
Howard Hughes Medical Institute (D.B.). This work
was also supported by the European Union's
Horizon 2020 research and innovation program
under grant agreement No. 681137 (M.C., R.W.S.).
C.A.C. is supported by a NIH F31 Ruth L.
Kirschstein Predoctoral Award AI131873 and by
the Achievement Rewards for College Scientists
Foundation. This work was partially funded by IAVI
with the generous support of USAID, Ministry of
Foreign Affairs of the Netherlands, and the Bill &
Melinda Gates Foundation; a full list of IAVI donors
is available at www.iavi.org. The contents of this
manuscript are the responsibility of the authors
and do not necessarily reflect the views of USAID
or the US Government. The funders had no role in
study design, data collection and analysis, decision
to publish, or preparation of the manuscript.

**Competing interests:** The authors have declared
that no competing interests exist.

HIV Env trimer immunogens; however, the optimal implementation to different immunization strategies remains to be determined.

## Author summary

Protein constructs based on soluble ectodomains of HIV glycoprotein (Env) trimers are the basis of many current HIV vaccine platforms. Multivalent antigen display is one strategy applied to improve the immunogenicity of various subunit vaccine candidates. Here, we describe and comprehensively evaluate a library of *de novo* designed protein nanoparticles of different geometries for their ability to present trimeric Env antigens. We found three nanoparticle candidates that can stably incorporate model Env trimers on their surfaces while maintaining structure and antigenicity. The designed nanoparticle immunogens had an increased capacity to stimulate B-cells expressing antigen-specific receptors. The immunogenicity of one nanoparticle candidate was assessed in rabbits. Nanoparticle presentation geometry appeared to alter the distribution of antibody responses against different epitopes while inducing similar serum binding titers and only slightly elevated neutralizing titers. In addition to introducing a novel set of reagents for multivalent display of Env trimers, this work provides both guiding principles and a detailed experimental roadmap for the generation, characterization, and optimization of Env-presenting, self-assembling nanoparticle immunogens.

## Introduction

Recombinant protein immunogens hold great promise against difficult viral and microbial targets for which there are currently no viable vaccine solutions (e.g. HIV, malaria). Well-defined structure, control over the exposure of different epitopes, high sample homogeneity and efficient manufacturing are some of the advantages of this vaccine platform [1]. Engineered ectodomains of the Env glycoprotein (Env) are at the core of most present HIV vaccine development efforts [2–12]. Recombinant native-like trimers, based on different HIV strains and carrying well-defined sets of stabilizing mutations, have been shown to elicit HIV-specific neutralizing antibody (NAb) responses in relevant animal models [13–16]. Several of these constructs are being evaluated in human clinical trials, with many others in pre-clinical testing stages [17,18] (ClinicalTrials.gov Identifiers: NCT03961438, NCT03816137, NCT03699241, NCT04046978).

While recombinant, native-like trimers represent a breakthrough in HIV vaccine development, identifying the most appropriate way to present them in order to maximize immunogenicity in different formulations is a major challenge that remains to be addressed. Interactions with elements of the innate and adaptive immune system are highly dependent on pathogen/immunogen shape and size, as well the distribution of surface antigens [1,19,20]. Multivalent particulate antigen presentation is important for several reasons. (1) An array of regularly spaced antigens can lead to strong, avidity-enhanced interactions with B-cell receptors (BCRs), resulting in more robust activation of antigen-specific B cells [21–25]. This factor is of particular importance for initial recruitment of immunologically naïve B cells with low affinity towards the antigen. (2) Multivalent presentation of glycosylated antigens such as HIV Env leads to more efficient crosslinking and opsonization by mannose-binding lectin (MBL), triggering the lectin pathway of the complement system [26,27]. The recruitment of complement

components facilitates recognition by antigen presenting cells (APC), enhancing uptake by dendritic cells (DC) and macrophages at the site of injection and by lymph node-resident DCs, resulting in better priming of effector T cells [28]. (3) Furthermore, antigen coating by MBL and complement components leads to improved trafficking through the layer of subcapsular sinus macrophages coating the lymph node, and greater accumulation of antigen within the lymph node follicles [26,27,29,30]. (4) Finally, particulate antigens with diameters in the 40–100 nm range require more time to penetrate the extracellular matrix around the site of injection to reach the lymphatic system, which results in slower release and hence prolonged antigen exposure [25,28,29,31,32]. Extended exposure to HIV Env antigens when osmotic pumps were used for controlled immunogen release was recently correlated with more robust B-cell responses, enhanced somatic hypermutation and a greater diversity of the elicited polyclonal antibody response [33].

Two-component, self-assembling nanoparticles are a versatile platform that can present HIV Env and other viral glycoproteins in a well-defined manner. [34–38]. A combination of symmetric protein modeling and RosettaDesign [39–42] is applied to generate particles of appropriate geometry consisting of two oligomeric building blocks, at least one of which is intended for genetic fusion to the antigen (antigen-bearing component). The second component is generally based on a different oligomeric protein scaffold and is essential for assembly, but it is typically not used for antigen presentation (assembly component). The two nanoparticle components can be purified independently, with assembly occurring when they are combined at an appropriate stoichiometric ratio. This strategy enables a high level of control over the structural and antigenic integrity of specific epitopes on the Env antigen [37,38]. In addition to achieving multivalent display, the geometry and spacing of presented antigens can also be varied through design [35,36,38]. This may be particularly important for vaccine design efforts focusing on a specific set of HIV Env epitopes, which need to be accessible in order to be immunogenic. For example, nanoparticles with high antigen density may be better suited for immunization focusing on the trimer apex. Conversely, if the epitope targets are closer to the Env-trimer base (e.g. the fusion peptide) then nanoparticles with greater antigen spacing must be applied to assure the accessibility of those epitopes. Flexibility of the system can also be influenced by adjusting the length and amino-acid composition of the linker connecting the antigen and the nanoparticle. The main advantage of the *de novo* nanoparticles platform is that they can be customized from the initial design stage to achieve the most optimal immune response against the antigen. Of course, this will also be antigen (and epitope) dependent and needs to be determined empirically.

Alternative methods for the multivalent presentation of HIV Env trimers include virus-like particles [43–45], liposomes [46–49]; synthetic nanoparticles [50–53]; and one-component nanoparticles based on natural protein scaffolds such as lumazine synthase [54,55], dihydrolipoylacetyltransferase [56] and ferritin [56–58]. Meta-analysis of the published immunization data [19] suggests that nanoparticle presentation of HIV Env trimer immunogens has not yet yielded strong improvements in immunogenicity as with human papillomavirus, respiratory syncytial virus (RSV) and influenza immunogens [34,59,60]. When comparing autologous neutralization titers between nanoparticle and free trimer immunogens, the significant improvement has only been observed for Envs with immunodominant neutralizing Ab epitopes located at the trimer apex (JRFL, ConM, 16055) [19]. Part of the explanation for the relatively modest benefits of nanoparticle presentation lies in the fact that HIV Env is an exceptionally challenging immunogen [61], but there is a number of other potentially contributing factors (nanoparticle instability in vivo, nanoparticle immunogenicity, choice of HIV Env antigen, incorporation of non-native Env trimers, location of immunogenic epitopes, occlusion of different epitopes by MBL etc.) [19,37,62,63]. More research is necessary to

understand the exact contribution from all of these various factors and select the most optimal nanoparticle platform for presentation of HIV Env immunogens. However, the existence of different complementary approaches provides a wealth of options for immunogen design with respect to geometry and antigen distribution.

Recently, a library of two-component nanoparticles of different symmetries (tetrahedral, octahedral and icosahedral) was designed *de novo* to support the display of viral glycoprotein antigens, and found to stably present RSV F, HIV Env, and influenza HA trimers [38]. Here, we perform a detailed characterization of the designed HIV Env nanoparticle immunogens, based on BG505 and consensus group M (ConM) Env constructs. The capacity to appropriately present BG505-SOSIP trimers [5,6] was assessed by testing the expression, assembly, structural and antigenic properties of the resulting nanoparticles and presented antigens. We then used rabbits to evaluate the immunogenicity of a particularly promising tetrahedral nanoparticle construct (T33_dn2) displaying the ConM-SOSIP trimer, in comparison to the same trimers delivered as soluble proteins.

## Results

### Nanoparticle library

A library consisting of five self-assembling nanoparticle candidates was designed for display of SOSIP-based trimeric ectodomains of HIV Env [38]. The library includes three tetrahedral (T33_dn2, T33_dn5 and T33_dn10), one octahedral (O43_dn18) and one icosahedral (I53_dn5) system (S1A Fig). The tetrahedral nanoparticles present 4 Env trimers when Env is fused to one of the nanoparticle components while their octahedral and icosahedral counterparts carry 8 and 20, respectively (S1B Fig). In each case, the antigen-bearing component (shown in orange in S1A Fig) was based on one of two computationally designed trimeric, helical repeat protein scaffolds– 1na0C3_2 [38] or 1na0C3_3 [64]–and its N terminus was genetically fused to the C terminus of the Env glycoprotein ectodomain. However, the amino acid sequence of each antigen-bearing component differs between the individual constructs (S1 Table), with the principal differences being in the residues comprising the computationally designed interface that drives nanoparticle assembly in the presence of the assembly component. The naming system used for *de novo* designed nanoparticles is illustrated in S1C Fig, using the example of I53_dn5 [38].

### Antigen-bearing component production and characterization

We selected a BG505-SOSIP trimer as the HIV Env model for initial optimization steps. The trimer was modified by incorporating the SOSIP.v5.2 and MD39 stabilizing mutations [3,5]. A further modification involved knocking in glycosylation sites at positions N241 and N289 to occlude the immunodominant glycan hole that is present in the BG505 trimer but generally rare in other HIV strains [16,65]. See the Methods section and S2 Table for details on engineered mutations. Together, these sequence changes raise the thermal stability of the trimer, increase its expression levels and may improve its ability to induce more broadly active NAbs by suppressing narrow-specificity responses [5,66]. This construct is hereafter referred to as BG505-SOSIP.v5.2(7S). Trimeric building blocks of the five nanoparticle candidates were genetically fused to the C terminus of BG505-SOSIP.v5.2(7S) to generate the various antigen-bearing components, which were expressed as secreted proteins in 293F cells. Compared to the unmodified BG505-SOSIP.v5.2(7S), all five of the engineered constructs were less efficiently expressed (Table 1 and S2A Fig). However, the four constructs that could be expressed and purified (see below) and the parental BG505-SOSIP.v5.2(7S) trimer all had very similar

**Table 1. Properties of BG505-SOSIP-fused nanoparticle components.**

| Construct | Protein yield after SEC (mg per 1L of 293F cells) | Tm (°C) | gp120 –gp41 cleavage | Nanoparticle assembly |
|---|---|---|---|---|
| BG505-SOSIP.v5.2(7S) | 1.5 ± 0.5 | 76.9 ± 0.5 | Full | - |
| BG505-SOSIP-T33_dn2A | 0.7 ± 0.2 | 77.9 ± 0.5 | Full | Yes |
| BG505-SOSIP-T33_dn5A | 1.0 ± 0.2 | 77.2 ± 0.5 | Full | No |
| BG505-SOSIP-T33_dn10A | 0.3 ± 0.1 | 77.5 ± 0.5 | Full | Yes |
| BG505-SOSIP-O43_dn18B | 0 | N/A | N/A | N/A |
| BG505-SOSIP-I53_dnB | 0.4 ± 0.1 | 77.7 ± 0.5 | Full | Yes |

melting temperatures suggesting that C-terminal fusion of 1na0C3_2/1na0C3_3-based constructs does not destabilize the SOSIP antigen (Table 1).

Among the antigen-bearing components evaluated, BG505-SOSIP-O43_dn18B had a high propensity towards self-aggregation, and no trimer could be recovered after purification by size-exclusion chromatography (SEC; S2A Fig). Although BG505-SOSIP-T33_dn5A expressed quite efficiently, it failed to co-assemble into a nanoparticle when mixed with the corresponding assembly component, T33_dn5B (Table 1, S2D Fig). These constructs were therefore not pursued further. BG505-SOSIP-T33_dn2A, -T33_dn10A and -I53_dn5B were expressed at ~20–50% of the BG505-SOSIP.v5.2(7S) trimer yield and efficiently assembled into nanoparticles (Table 1, S2D Fig). The yield reductions were probably due to self-aggregation, as indicated by SEC profiles (S2A Fig). The absence of appropriate assembly component during expression leads to solvent exposure of nanoparticle interface residues in the antigen-bearing component that are mainly hydrophobic, which is the most probable cause of the observed self-aggregation. The above three candidates were then further characterized. An SDS-PAGE analysis showed that the SEC-purified samples were homogeneous and cleaved into gp120 and gp41 (+ nanoparticle component) subunits that were separated when a reducing agent was added to break the engineered disulfide bonds (Fig 1A). Analysis of 2D class-averages from negative stain electron microscopy (NS-EM) imaging suggests that the BG505-SOSIP trimer components are in a native-like conformation, with the density corresponding to the fused nanoparticle component clearly discernible (Fig 1B; see red arrows). A low level (~ 8%) of monomeric protein was present only in the BG505-SOSIP-T33_dn10A sample.

We used Biolayer Interferometry (BLI) and a panel of antibodies (IgG) specific to different Env epitopes and conformations to probe the antigenicity of the antigen-bearing components in their trimeric forms, prior to nanoparticle assembly (Fig 1C, S2C Fig). All the samples interacted with neutralizing antibodies (NAbs) specific to the closed, prefusion state of the BG505-SOSIP trimer (VRC34, PGT145, PGT151, 2G12, 3BC315), but did not bind to b6, 19b, F105 and 14e, non-neutralizing antibodies (non-NAbs) that recognize more open trimer conformations and free monomers. The RM19R and RM20A3 antibodies showed markedly lower binding to the BG505-SOSIP-T33_dn10A and -I53_dn5B components compared to the BG505-SOSIP.v5.2(7S) trimer. These two non-NAbs interact with the base of the BG505-SOSIP.v5.2(7S) trimer, which is the site of fusion to nanoparticle component. The occlusion of these epitopes on the antigen-bearing fusion constructs is expected and potentially useful, as the induction of non-NAbs against the trimer base could be immune-distractive. Surprisingly, BG505-SOSIP-T33_dn2A displayed no change in RM19R binding compared to free BG505-SOSIP.v5.2(7S), and only a marginal decrease in binding to RM20A3. The different behavior of BG505-SOSIP-T33_dn2A may be explained by the fact that T33_dn2A scaffold is constructed from a different trimeric component (1na0C3_3; [64]) and uses a different linker

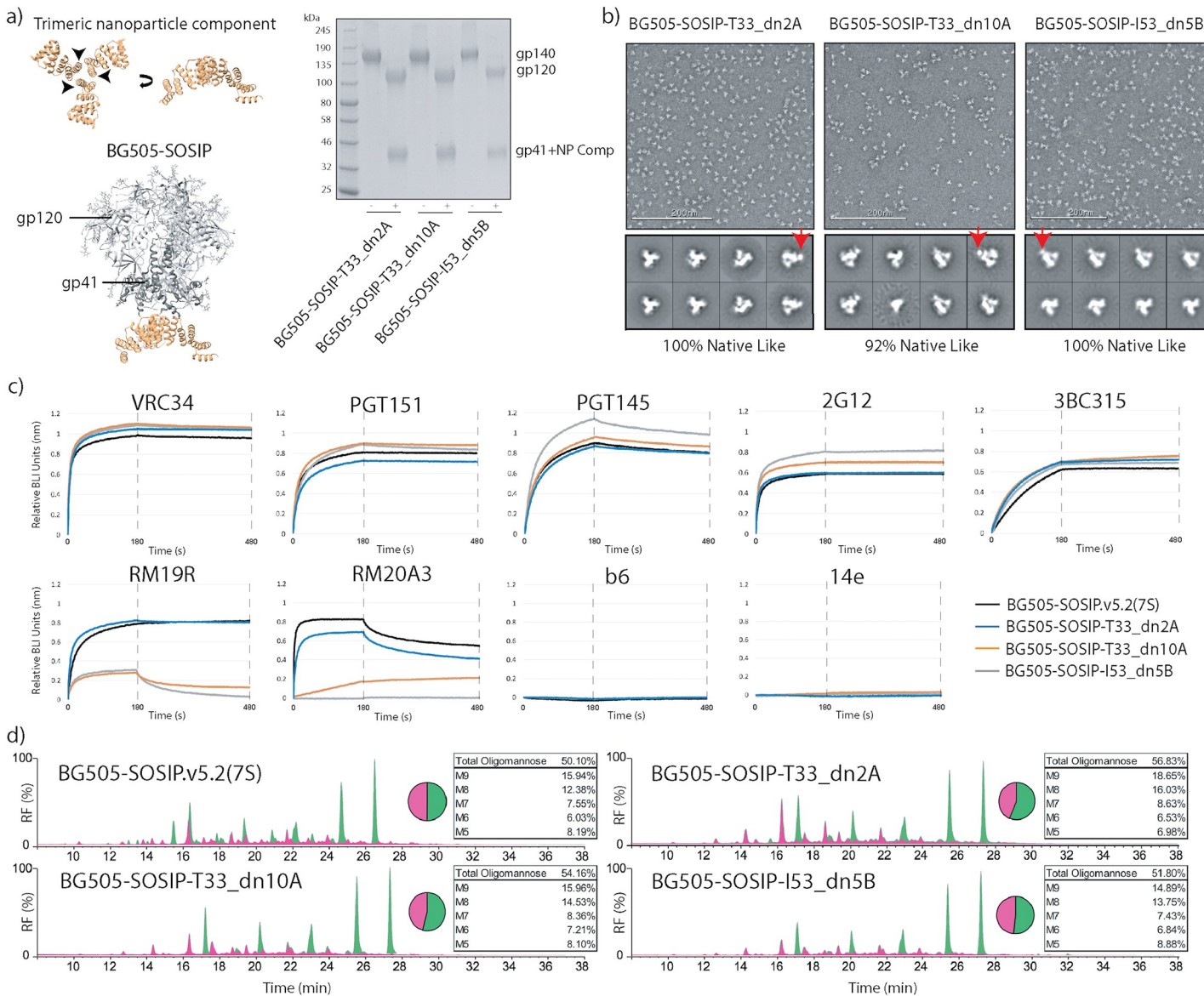

**Fig 1. Evaluation of the antigen-presenting components.** (a) Antigen-bearing components were generated by fusing the N-termini of trimeric nanoparticle building blocks to BG505-SOSIP.v5.2(7S) (*left*). The SDS-PAGE gel of the purified antigen-bearing components denatured in the presence (+) and absence (-) of reducing agent (right). (b) NS-EM analysis of the purified antigen-bearing components (representative raw micrograph and 2D class averages). Red arrows indicate the location of the fused nanoparticle component. Analysis of the resulting 2D classes suggests that the Env antigen is in a native-like, trimeric form in all three antigen-bearing components. A small percentage of monomer/dimer-like particles were detected in the BG505-SOSIP-T33_dn10A sample (2nd class from the left in the bottom row). (c) BLI analysis of the antigenicity of three antigen-bearing components compared to BG505-SOSIP.v5.2(7S). (d) Glycan composition analysis for the three antigen-bearing components and BG505-SOSIP.v5.2(7S). Peaks sensitive to endoglycosidase H digestion correspond to oligomannose-type glycans and are colored green.

than BG505-SOSIP-T33_dn10A and -I53_dn5B, which comprise highly related trimeric components (1na0C3_2; [38]) and identical linkers.

In a glycan composition analysis, the ratio of oligomannose and complex glycans was very similar for BG505-SOSIP-T33_dn2A, -T33_dn10A, -I53_dn5B and the parent BG505-SOSIP. v5.2(7S) trimer, with the oligomannose proportion ranging from 50% to 57% (Fig 1D). The ratio of individual oligomannose species remained similar in all samples, with the highest

difference being present in BG505-SOSIP-T33_dn2A, where the relative content of oligomannose species of higher molecular weight (M8 and M9) was slightly elevated.

Site-specific glycan composition analyses showed that glycan processing was conserved at key epitopes on all the trimer samples, with sites such as N332 and N160 containing predominantly oligomannose-type glycans (S3 Fig). One major difference was, however, visible at gp41-site N637 on BG505-SOSIP-I53_dn5B, which contains 100% oligomannose-type glycans compared to 34% on BG505-SOSIP.v5.2(7S). Additionally, this site is fully glycosylated in the three antigen-bearing components, and only partially glycosylated (56%) on the BG505-SO-SIP.v5.2(7S) trimer. Several glycan sites on gp120 of the BG505-SOSIP-I53_dn5B and -T33_dn10A components were also enriched for oligomannose-type glycans, most notably N276 and N355. Specifically, the oligomannose contents of 67% for N276 and 22% for N355 on BG505-SOSIP.v5.2(7S) increased to >95% for N276 and >73% for N355 on the BG505-SOSIP-I53_dn5B and -T33_dn10A components. The V1/V2 glycan sites N133, N137 and N185e were <95% occupied on all the nanoparticle fusion proteins, while the N156 site on BG505-SOSIP.v5.2(7S) was 89% occupied (in the antigen-bearing components the occupancy at N156 is >99%). Among the closely spaced glycosylation sites at N289, N295 and N301, N289 and to a lesser extent N301, glycosylation appeared to be affected by the fusion to the antigen-bearing component of the nanoparticle. A reduction of 13–19% was observed in the occupancy at N289 in the three tested antigen-bearing components when compared to the parental BG505-SOSIP.v5.2(7S).

Collectively, these data indicate that several of the *de novo*-designed nanoparticle trimers were able to present a genetically fused native-like Env trimer without major changes to antigen structure, stability, or glycan profile.

## Nanoparticle assembly, antigenic and structural characterization

The three antigen-bearing components described above (BG505-SOSIP-T33_dn2A, -T33_dn10A and -I53_dn5B) were then tested for nanoparticle assembly (Fig 2A). The corresponding assembly components required for nanoparticle formation, T33_dn2B, T33_dn10B and I53_dn5A, were expressed in *E. coli* and purified as described in the Methods section. Analysis of the purified assembly components by SDS-PAGE is shown in S2B Fig. Equimolar amounts of the antigen-bearing component and the corresponding assembly component (on a subunit:subunit basis) were combined and incubated for 24 hours at three different temperatures (4, 25 and 37°C) before assembly was evaluated using native PAGE. In each case, assembly efficiency increased with the incubation temperature. The tetrahedral nanoparticles, T33_dn2 and T33_dn10, assembled at a high yield (~80–100%) under all the conditions tested. However, the icosahedral nanoparticle, I53_dn5, assembled less efficiently. After a 24-hour incubation at 37°C, only ~30% of the input material migrated as nanoparticles on a native gel, though when the incubation period was extended to 72 h at 37°C, the yield increased to ~70% (S2D Fig).

The presence of both the antigen-bearing component and the assembly component in SEC-purified nanoparticles was verified using SDS-PAGE (S2E Fig). Sample homogeneity and structural integrity were assessed using NS-EM. Representative raw micrographs, 2D class-averages and reconstructed 3D models (adapted from [38]) show that the assembled forms of the nanoparticles were consistent with the predictions of the computational design models, with individual building blocks clearly discernible in each case (Fig 2B).

Nanoparticle stability under various conditions was evaluated using a native PAGE assay (S4 Fig). The tetrahedral particles (T33_dn2 and T33_dn10) were highly stable in buffers with pH values in the range 5–9 and at NaCl concentrations from 25–1000 mM. They also remained assembled at temperatures up to 65°C and withstood multiple freeze-thaw cycles. In

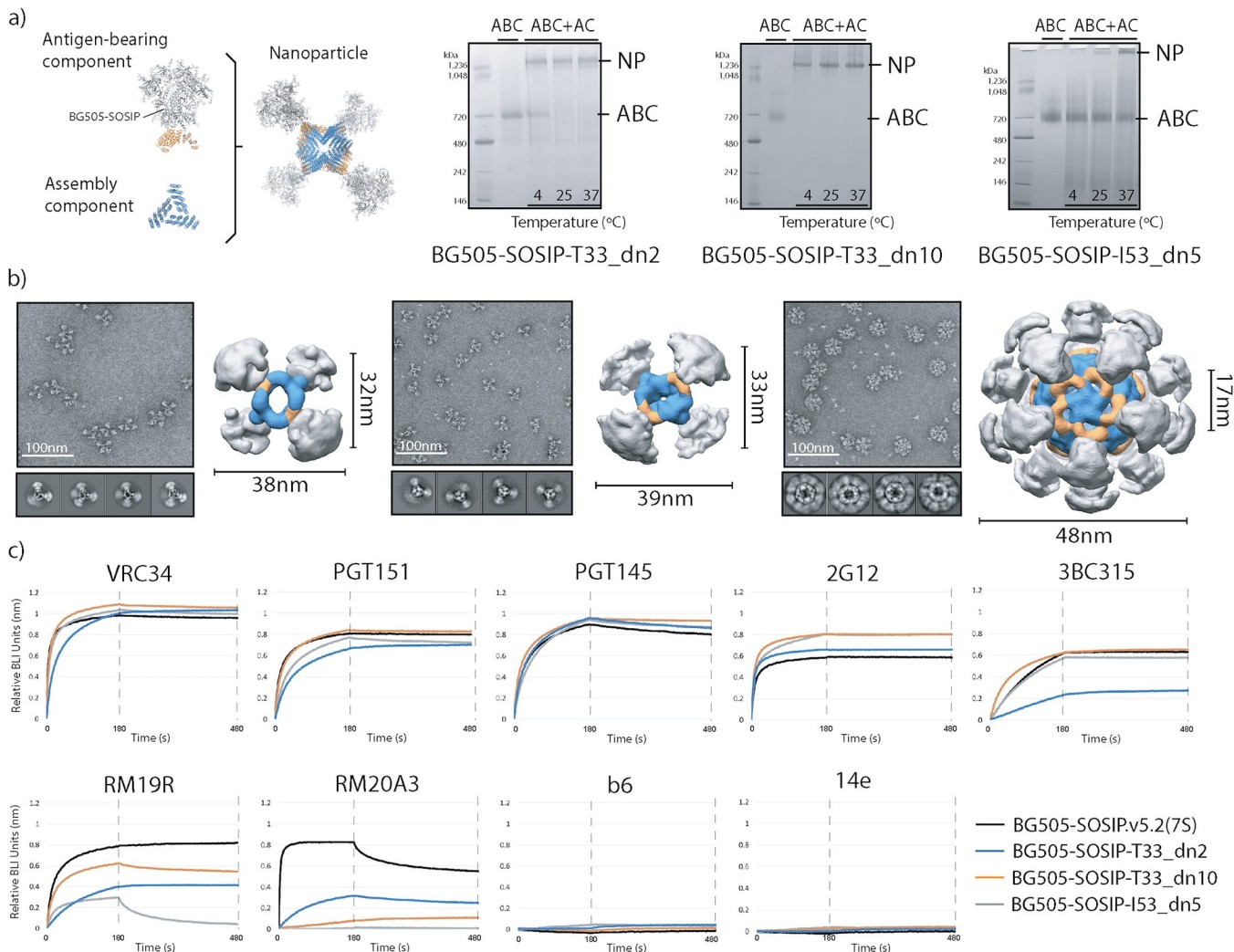

**Fig 2. Nanoparticle assembly and characterization.** (a) Schematic representation of individual components and assembled nanoparticle (left). Nanoparticle assembly tests were performed at different temperatures and the assembly efficacy was assessed using Native PAGE. NP, nanoparticle; ABC, antigen-bearing component; AC, assembly component; (right). (b) Negative stain EM analysis of purified nanoparticles. Representative raw micrographs, 2D class averages and 3D reconstructions are shown for BG505-SOSIP-T33_dn2 (left), -T33_dn10 (middle) and -I53_dn5 (right) nanoparticles. 3D maps are segmented and color-coded (BG505 SOSIP, gray; antigen-bearing component, orange; assembly component, blue). Particle diameters and the average apex-apex distance between the two closest neighboring Env trimers are shown for each nanoparticle. These data are also described in Ueda, Antanasijevic et al., 2020 [38]. (c) BLI analysis of antigenicity of assembled nanoparticles compared to the BG505-SOSIP.v5.2(7S) trimer.

contrast, the icosahedral nanoparticles were less stable under the various test conditions and were particularly sensitive to the freeze-thaw procedure. The latter observations are consistent with the presence of significant amounts of unassembled components revealed by NS-EM (Fig 2B, right), and also with the greater difficulties in assembling the icosahedral particles (see above).

We used BLI to assess whether nanoparticle presentation interfered with the accessibility of antibody epitopes, compared to the parental BG505-SOSIP.v5.2(7S) trimer (Fig 2C, S2F Fig). Overall, the antigenic profiles of the assembled nanoparticles, the corresponding antigen-bearing components and the free trimer were very similar (Fig 2C, and see also Fig 1C). Compared to the BG505-SOSIP.v5.2(7S), the greatest decrease in binding was seen with the base-specific antibodies, RM19R and RM20A3, suggesting that nanoparticle assembly further decreases the

accessibility of the trimer base (Fig 2C). There was also a decrease in the binding of the BG505-SOSIP-T33_dn2 nanoparticles to the 3BC315 broadly neutralizing antibody (bNAb), compared to the free trimer (Fig 2C). As no such difference was seen with the BG505-SO-SIP-T33_dn2A antigen-bearing component (Fig 1C), assembly of this nanoparticle appears to reduce the accessibility of epitopes for bNAbs such as 3BC315 that are located near the bottom of the trimer.

## Cryo-EM analysis of assembled nanoparticles

We used cryo-electron microscopy (cryo-EM) to obtain more detailed information on the structures of the BG505-SOSIP-T33_dn10 and -I53_dn5 nanoparticles and the Env trimers they display (Fig 3). Cryo-EM characterization of the BG505-SOSIP-T33_dn2 nanoparticle has been performed previously and reported elsewhere [67]. The data were processed as described in the Methods section and summarized in S5 Fig and S6 Fig, with data acquisition and processing statistics shown in S3 Table. Due to the flexible nature of the linker between the nanoparticle component and the BG505-SOSIP trimer, the initial 3D reconstructions of the complete nanoparticles generated only highly diffuse and poorly defined density for BG505-SOSIP (Fig 3, S5 Fig and S6 Fig). The data were therefore computationally segmented to reconstruct the nanoparticle core and the displayed trimer as two independent but flexibly linked entities (sub-particles).

Nanoparticle reconstruction was performed using a focused refinement procedure in which a solvent mask around the nanoparticle core excluded the signal originating from BG505-SOSIP trimers. Symmetry was applied in all 3D classification and refinement steps: tetrahedral for T33_dn10 and icosahedral for I53_dn5. The final resolutions of the reconstructed maps were 4.25 Å and 12.50 Å for T33_dn10 and I53_dn5, respectively. The T33_dn10 core design model from Rosetta Design was relaxed into the EM map using a combination of Rosetta relaxed refinement [68] and manual refinement in Coot [69]. Model refinement statistics are shown in S4 Table. The model-to-map fit for the refined structure is shown in Fig 3A (right). There are only small, local structural differences between the Rosetta-predicted, unliganded [38] and BG505-SOSIP-bearing models of the T33_dn10 nanoparticle core. The Cα RMSD between the experimentally determined and predicted models is 1.43 Å (on the level of the asymmetric unit), demonstrating that the presence of four displayed BG505-SOSIP trimers did not cause any major structural rearrangements within the nanoparticle core. Model refinement was not performed for the I53_dn5 core because the resolution was too poor. The design model of I53_dn5 was, however, highly concordant with the reconstructed EM map (Fig 3B, right).

The nanoparticle-displayed BG505-SOSIP trimers were reconstructed using a localized reconstruction approach [70]. The signal corresponding to the nanoparticle core was sub-tracted from the particle images and the trimer sub-particles were extracted and aligned independently. The final BG505-SOSIP trimer subset from the T33_dn10 nanoparticle dataset consisted of 84,435 sub-particles and was reconstructed to 4.14 Å resolution. A model of the displayed BG505-SOSIP was refined using a combination of Rosetta-relaxed and manual refinement in Coot (Fig 3A, S4 Table). The close agreement of the refined model with pub-lished structures of the BG505-SOSIP trimer (PDB entries: 5CEZ and 5ACO [71,72]) implies that nanoparticle assembly does not interfere with the structural integrity of the trimer. For the I53_dn5 nanoparticle dataset, the final subset included 7,737 sub-particles and was recon-structed to a resolution of 6.67 Å. The quality of the BG505-SOSIP map was significantly better than the corresponding I53_dn5 nanoparticle core but still too poor resolution to refine a structural model. However, a published structure (PDB entry: 5CEZ [71]) was docked into the map and exhibited a good fit (Fig 3B).

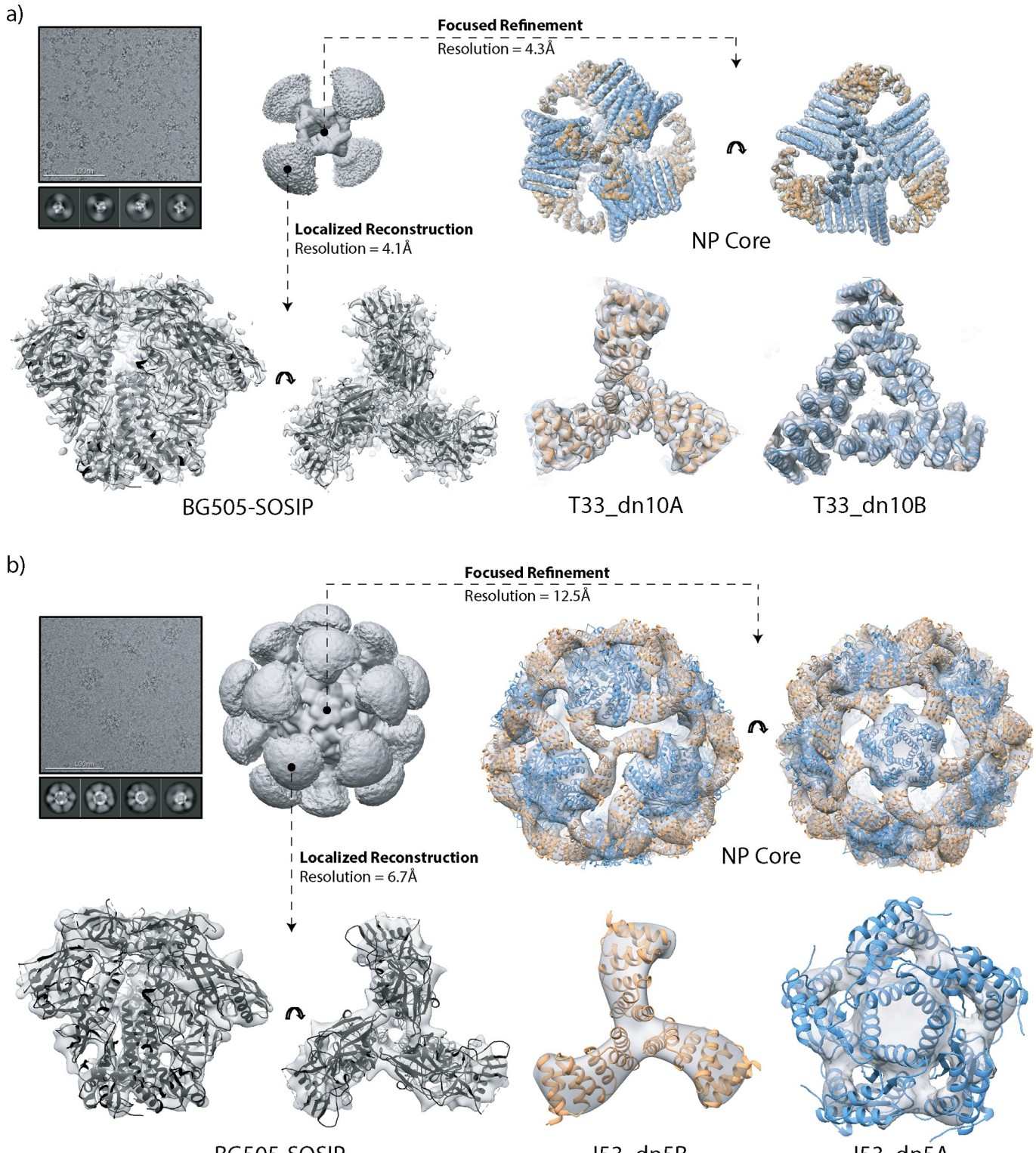

**Fig 3. Cryo-EM analysis of tetrahedral and icosahedral nanoparticles.** (a) Tetrahedral BG505-SOSIP-T33_dn10 nanoparticle; (b) Icosahedral BG505-SOSIP-I53_dn5 nanoparticle. Sample micrograph, 2D class averages and initial 3D reconstructions of the full nanoparticles are displayed in the top left part of the corresponding panels. Focused refinement was applied to generate a 3D reconstruction of the nanoparticle core (top and bottom right, maps are in light gray). The refined model of T33_dn10 and the Rosetta_design model of I53_dn5 are docked into the corresponding maps (antigen-bearing component, orange; assembly component, blue). Localized reconstruction approach was used for analysis of the presented antigen (bottom left). Refined BG505-SOSIP models are shown in black.

## Nanoparticle presentation affects trimer interaction with antigen-specific B cells

BG505-SOSIP-T33_dn2, -T33_dn10, and -I53_dn5 nanoparticles were then functionally analyzed as immunogens. Their capacity to stimulate antigen-specific B cells was evaluated using K46 mouse B-cell lines that expressed IgM versions of three different HIV-specific bNAbs on their surfaces (PGT145, VRC01 and PGT121; Fig 4). The B cells were treated with equimolar amounts of BG505-SOSIP.v5.2(7S) antigen presented as free trimers, antigen-bearing components or assembled nanoparticles, and the relative number of cells responding to each antigen was quantified (Fig 4). $Ca^{2+}$ mobilization inside B cells, measured by fluorescence-activated cell sorting (FACS), was used as an indicator of antigen-induced activation. For clarity, the data from each B cell line are presented in two panels, the first showing the different antigen-bearing trimeric components, and the second the assembled nanoparticles. The free trimers (unmodified BG505-SOSIP.v5.2(7S) and the antigen-bearing components) induced only very low levels of $Ca^{2+}$ flux (Fig 4, left column) suggesting that the local antigen concentration was insufficient for efficient BCR crosslinking and B-cell activation [73]. In contrast, presenting the same trimers on the surface of each of the three nanoparticles activated the B cells much more strongly, as quantified by the higher percentage of cells in which a $Ca^{2+}$ flux occurred (Fig 4, right column). The icosahedral BG505-SOSIP-I53_dn5 nanoparticle triggered stronger $Ca^{2+}$ signal than the two tetrahedral nanoparticles in B cells expressing PGT145. In VRC01-expressing cells this difference was less pronounced, particularly when compared to BG505-SOSIP-T33_dn10. Although the signal-to-noise ratio was generally low in PGT121-expressing cells, there was an increased $Ca^{2+}$ flux response with the nanoparticles compared to free trimers.

## Generation and characterization of ConM-SOSIP-T33_dn2 nanoparticles

In the next step we wanted to evaluate the immunogenicity of nanoparticle-presented Env trimers. For this purpose, we selected the T33_dn2 nanoparticle platform. Despite the somewhat lower capacity to activate antigen-specific B-cells (compared to the I53_dn5 and T33_dn10), this was the most optimal system from the standpoint of production (highest yield) and stability. Additionally, these nanoparticles have previously been used for *in vivo* immunogen trafficking experiments in non-human primates [67], where we have demonstrated that BG505-SOSIP-T33_dn2 nanoparticles, unlike free trimers, get efficiently transferred through the layer of subcapsular macrophages and accumulate within lymph node follicles. This allowed us to test if improved lymph node localization results in superior antibody response against the presented trimer antigen. Finally, with only 4 trimers per particle, this nanoparticle has the lowest valency compared to other nanoparticle systems used in the field for the presentation of HIV Env trimers. Therefore, it would be of interest to assess the immunogenicity of tetrahedral nanoparticle systems and compare to other previously reported nanoparticle platforms [37].

Immunization experiments were performed with consensus group M (ConM) HIV Env trimers instead of BG505. This change was introduced for several reasons. Firstly, we wanted to evaluate if the nanoparticle platform described here can be used to present SOSIP trimers based on another HIV Env sequence. Also, ConM-SOSIP trimers have been shown to reproducibly elicit strong neutralizing antibody response against the V1/V2 loops at the trimer apex, which serves as an excellent readout when comparing different immunogens [8]. Finally, the immunogenicity of ConM-SOSIP on octahedral ferritin and icosahedral I53-50 nanoparticles presenting 8 and 20 copies of the trimeric antigen on the surface has been recently described [37]. These studies and ours were contemporaneous and were carried out under the

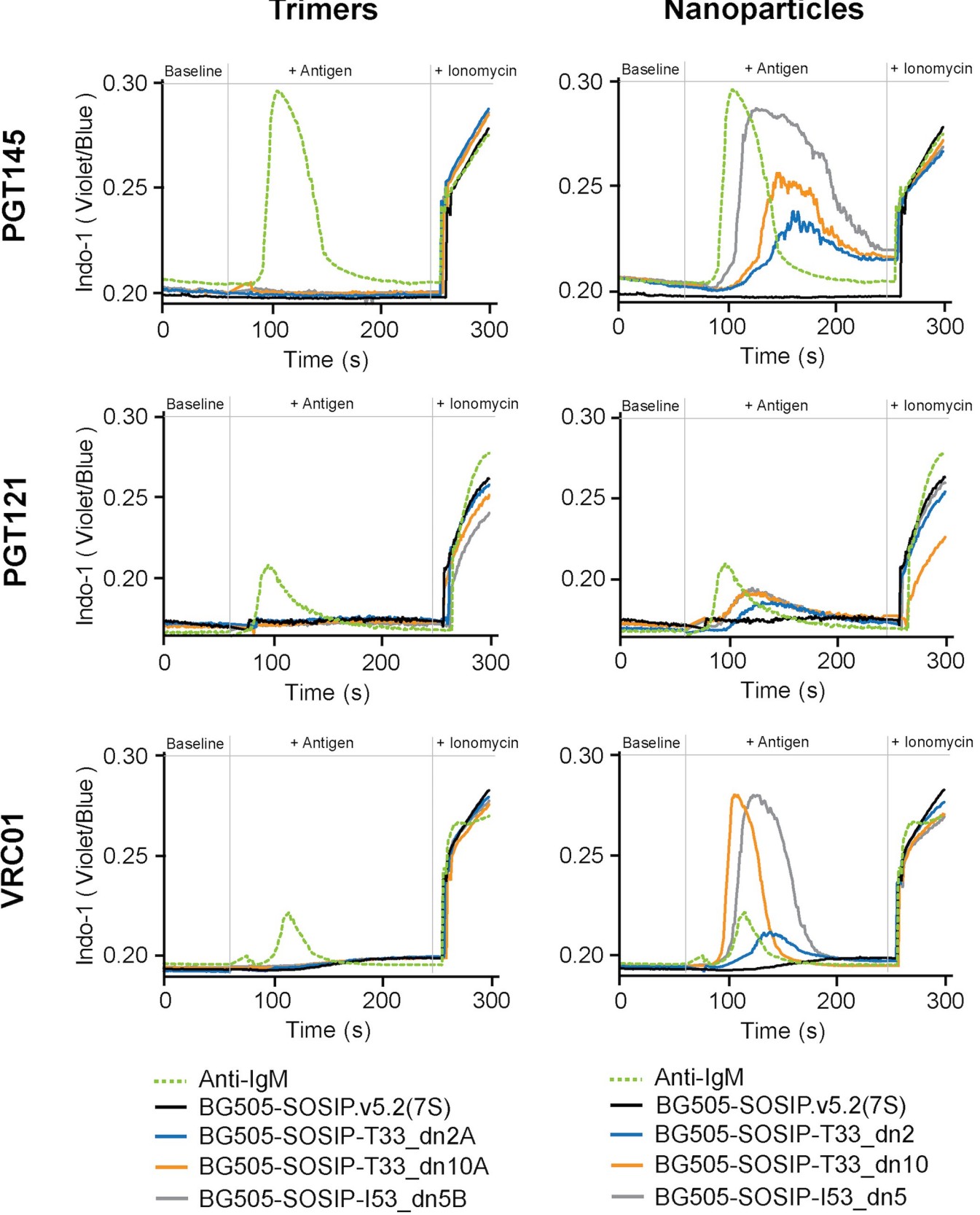

**Fig 4. B-cell activation by trimeric components and nanoparticles.** $Ca^{2+}$ flux (Indo I fluorescence) was used to assess the activation of B cells expressing HIV Env-specific IgM receptors (PGT145, VRC01 and PGT121) by equimolar amounts of BG505-SOSIP in the form of free trimers, fused to the antigen-bearing components (left column) and on the surface of assembled nanoparticles (right column). The antigen was introduced 60 s after the start of each measurement. Ionomycin was added after 240 s. Anti-IgM antibody was used as a positive control.

same conditions (animal model, dose, immunization regimen, adjuvant) to permit a comparison of data derived using immunogens of different valencies. Importantly, similar to BG505-SOSIP, ConM-SOSIP trimers are being evaluated clinically (ClinicalTrials.gov Identifiers: NCT03961438, NCT03816137), and are a highly relevant reagent for HIV vaccine design.

The ConM-SOSIP construct used in the experiments, termed ConM-SOSIP.v7, was engineered to include the SOSIP.v5.2 and TD8 stabilizing mutations (see the Method section and S1 Table for sequence information) [8,37,74]. The trimeric ConM-SOSIP-T33_dn2A component expressed at a similar level as its BG505-based counterpart (~ 0.6 mg per 1 L of 293F cells after SEC, S7A Fig), and assembled efficiently into nanoparticles when incubated with equimolar amounts of the T33_dn2B component at 4°C (> 90% assembly after 24 h, S7B Fig). The purified ConM-SOSIP-presenting nanoparticles were characterized by NS-EM and Surface Plasmon Resonance (SPR) (S7B, S7C and S7D Fig). The NS-EM analysis showed that the ConM-SOSIP-T33_dn2 nanoparticles assembled to the target tetrahedral architecture and were highly homogeneous. As assessed by SPR, the ConM-SOSIP.v7 trimers on their surfaces retained the capacity to interact with trimer-specific and other bNAbs. An antigenicity comparison of T33_dn2 nanoparticles with the free ConM-SOSIP.v7 trimer at equimolar amounts of Env per volume suggested that the nanoparticles efficiently presented NAb epitopes located on the upper half of the trimer but that the accessibility of epitopes located towards the bottom of the trimer was partially impaired. The stronger signals for binding to the apical, CD4bs, NAb epitopes reflects the 5.1-fold greater mass of the particles (S7C Fig). For the interface, fusion-peptide, and interprotomeric-gp41 epitopes the relative nanoparticle to trimer binding ranked 35O22, ACS202, PGT151, VRC34 (N123), RM20F, followed by 3BC315; thus already the relative nanoparticle to trimer binding to 35O22 was reduced compared with the binding to the upper epitopes, and the coincidental near superimposition of the two VRC34 (N123) curves reflects a substantially reduced epitope accessibility on the nanoparticles. Additional SPR analyses with the converse approach of immobilized trimer and nanoparticle (S7D Fig) showed low or absent binding to non-NAb, V3 and the CD4-bs epitopes on both trimer and nanoparticles, whereas the stoichiometry of maximum binding to bNAb epitopes was identical or nearly so on trimer and nanoparticles. These data are consistent with the similar SPR-based epitope accessibility studies performed on T33_dn2 nanoparticles presenting BG505-SOSIP.v5.2(7S): the accessibility of the epitopes towards the base is reduced but less so than on icosahedral nanoparticles [38].

## Immunogenicity of ConM-SOSIP-T33_dn2 nanoparticles

The ConM-SOSIP-T33_dn2 nanoparticles were then tested for immunogenicity in New Zealand White Rabbits and compared to the soluble ConM-SOSIP.v7 trimer (Fig 5). Two groups of 5 rabbits were used, with the immunogen dose adjusted to ensure that all the animals received an equimolar amount of ConM-SOSIP.v7 trimer (30 μg). The rabbits were immunized at weeks 0, 4 and 20, with blood draws performed at weeks 0, 2, 4, 6, 8, 12, 16, 20 and 22. Immune responses were monitored in sera by measuring ConM-SOSIP trimer-specific antibody titers, by ELISA, and NAb titers using the TZM-bl cell assay.

Both immunogens induced anti-ConM-SOSIP binding antibodies (Fig 5, middle panels; S5 Table). The mean ELISA binding titers were comparable between the immunization groups at

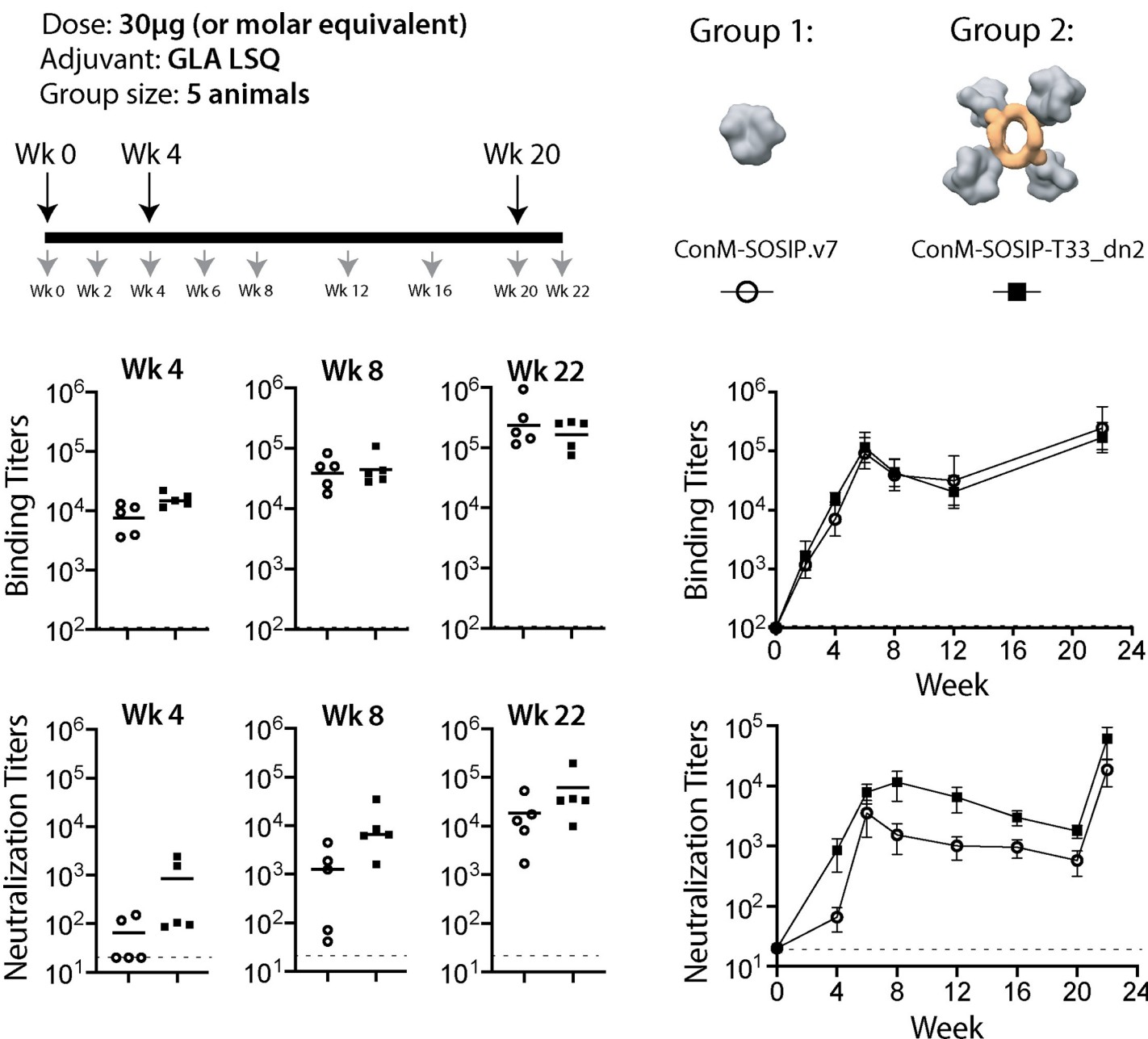

**Fig 5. Immunogenicity of nanoparticle-presented ConM-SOSIP.** Two groups of 5 rabbits were immunized with 30 μg of soluble ConM-SOSIP.v7 trimer (Group 1, open circles) or the equivalent amount presented on the T33_dn2 tetrahedral nanoparticle (Group 2, black squares). The immunization (large arrows) and bleed (small arrows) schedules are shown on the top left, and depictions of the immunogens on the top right. The serum anti-trimer binding and neutralizing antibody titers in each animal at weeks 4, 8 and 22 are presented as scatter plots with the mean titers indicated by lines. The mean titers for each group are plotted longitudinally on the right with the bars corresponding to the standard error of the mean. An AUC statistical analysis (based on two-tailed Mann-Whitney U-test) of titer values in Group 1 versus Group 2 as a function of time results in p values of 0.69 and 0.056 for the binding and neutralization titers, respectively. The data for Group 1 (ConM-SOSIP.v7) were adapted from Brouwer et al., 2019 ([37]; see Methods section for details).

each time point. An area under the curve (AUC) statistical analysis using a two-tailed Mann-Whitney U-test generated a p-value of 0.69, implying that there is no statistically significant difference in the binding titers between the two groups across all time points. Antibody responses of similar magnitude (week-22 ELISA titers of ~$10^5$) against the nanoparticle core

were also detected in the ConM-SOSIP-T33_dn2 immunogen group but not the soluble trimer group (S8A Fig).

NAb titers against the autologous Tier-1 ConM virus were more variable than the binding antibody responses, both within and between the immunization groups (Fig 5, bottom panels; S6 Table). Within each group, the NAb titers spanned a range of ~100-fold at each time point, a greater variation than the ~10-fold range in the anti-trimer titers. Every rabbit immunized with ConM-SOSIP-T33_dn2 nanoparticles generated a detectable NAb response (titer >20) after the first immunization, and the mean titers were consistently higher in this group than in the ConM-SOSIP.v7 group (p = 0.016 by AUC analysis) (S8B Fig). Taken together, the serological data suggest that presenting the ConM-SOSIP.v7 trimer on the T33_dn2 nanoparticle increased the proportion of the B-cell response directed against the autologous NAb epitope(s).

Next we tested if the sera from immunized rabbits (collected at week 22) can neutralize heterologous viruses pseudotyped with Tier-1 and Tier-2 HIV Envs. There was no detectable neutralizing response against the tested Tier-2 viruses (S7 Table). This finding is consistent with the previous immunization experiments with ConM-SOSIP.v7 trimers [8,37]. Cross-neutralization was the highest (ID$_{50}$ values in the $10^2$–$10^3$ range) with ConS-based virus (S8 Table). This is not surprising given that ConS is a Tier-1B consensus sequence Env, closely related to ConM. NAb titers against the tested Tier-1A viruses (SF162, MW965 and MN.3) were relatively low and similar between the animals in two groups (trimer and nanoparticle). High NAb titers against these viruses is often indicative of the presence of non-NAbs (for Tier 2 Envs) targeting the V3-tip, a family of epitopes that is inaccessible on native-like trimers. Altogether, the data suggest that the neutralizing antibody response elicited in both groups of animals is immunogen-specific and not very cross-reactive.

Finally, we sought to compare the immunogenicity of the tetrahedral nanoparticle platform described here (T33_dn2) against the nanoparticles of higher valency. As previously mentioned, this study was performed in parallel to the immunization experiments with ConM-SOSIP-ferritin (octahedral, 8 trimers per particle) and ConM-SOSIP-I53-50 (icosahedral, 20 trimers per particle) nanoparticles (described in [37]) and under the equivalent experimental conditions (animal model, immunogen dose, adjuvant, immunization regiment). This allowed us to directly compare the immunization data between different groups of animals (S9 Fig). First, we looked at the autologous neutralization titers (ID$_{50}$). Animals receiving ConM-SOSIP-T33_dn2 immunogen (Grp 2) developed neutralizing responses of similar magnitude against the ConM virus as Grp 3 (immunized with ConM-SOSIP-ferritin) and Grp 4 (immunized with ConM-SOSIP-I53-50); no statistically significant difference (with respect to Grp 2) was observed at any of the time points. When comparing the heterologous neutralization against Tier-1 pseudotyped viruses at the week-22 time point, we observe that animals receiving the tetrahedral and icosahedral nanoparticle immunogens (Grps 2 and 4), developed relatively low neutralizing responses compared to animals receiving ConM-SOSIP-ferritin (Grp 3). Ferritin, unlike T33_dn2 and I53-50, is a one-component nanoparticle that assembles in cells during expression. This can lead to incorporation of a certain percentage of non-native-like trimers into the assembled particle and elicitation of stronger anti-V3-tip response upon immunization. Altogether, the tetrahedral nanoparticle, T33_dn2, despite the modest valency of 4 trimers per particle, displays comparable immunogenic properties to the higher valency nanoparticles ferritin and I53-50.

## EM-based polyclonal epitope mapping of the elicited polyclonal antibody responses

We next employed EM-based polyclonal epitope mapping (EMPEM) [75] to characterize the specificities of antibodies elicited by the two immunogens (Fig 6). Polyclonal Fab samples

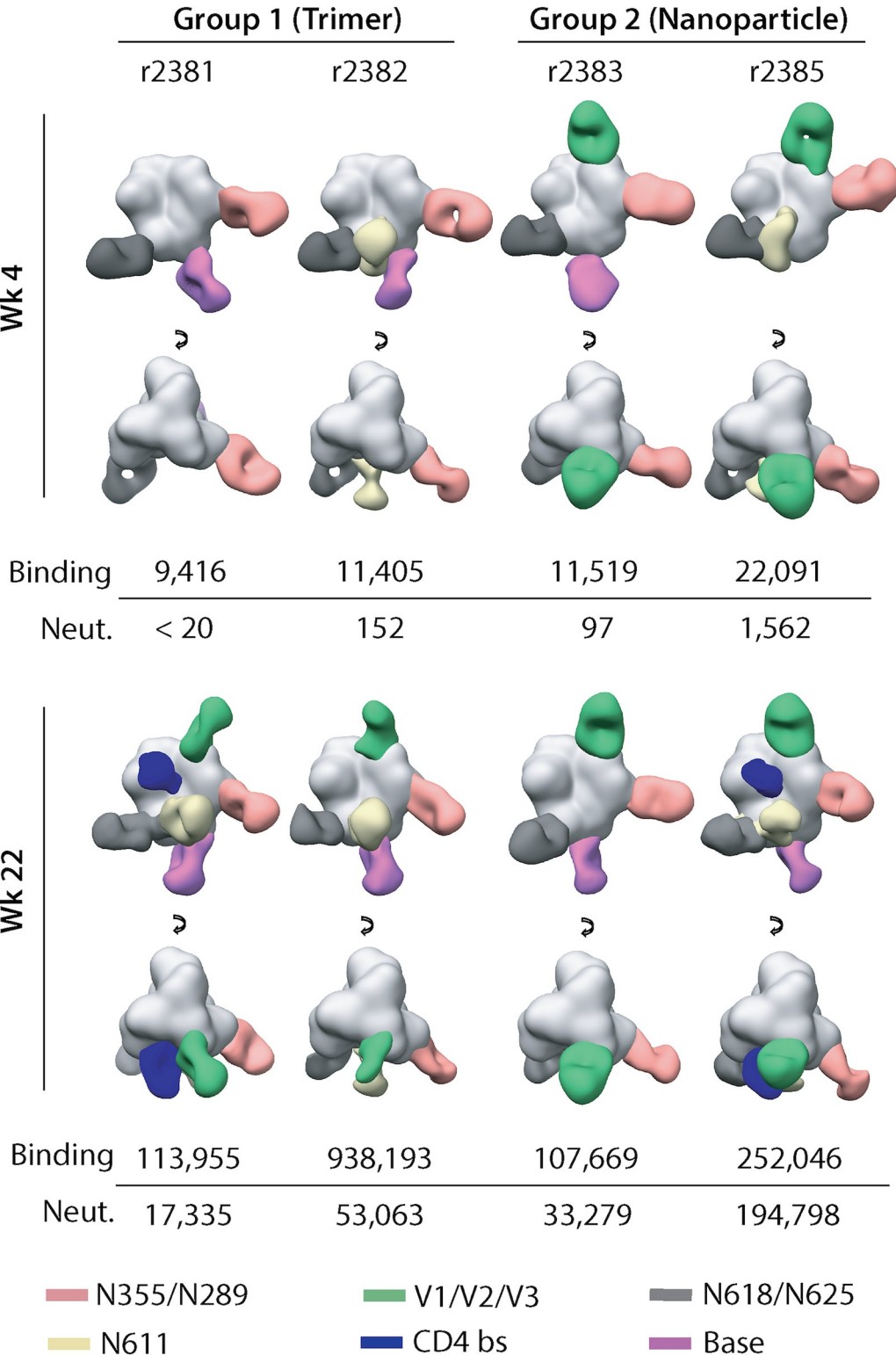

**Fig 6. EMPEM analysis of antibody responses in immunized animals.** Composite figures generated from EMPEM analysis performed using sera collected from the two animals in each group that have the highest ConM NAb titers at week 22. Data are shown for the week 4 (post-prime) and week 22 (post-final boost) time points. For simplicity, only a single antibody is shown for each epitope cluster. Epitope definitions are summarized in the text and color coded as indicated at the foot of the figure. The anti-trimer binding antibody and neutralizing antibody titers for each serum sample are listed below the images.

were prepared from week-22 sera from the two animals in each group with the highest ConM NAb titers (Group 1, r2381 and r2382; Group 2, r2383 and r2385), and also from the same animals at week 4. Given that the experiments were performed with samples from 2 animals from each group, the conclusions may not be generally applicable to the group as a whole. The purified Fabs were complexed with ConM-SOSIP.v9 trimers, which contain additional stabilizing mutations (see Methods and S10 Fig). Raw data, sample micrographs, 2D class averages and reconstructed 3D maps of the trimer-Fab complexes are shown in S11 Fig. Composite figures based on these datasets are shown in Fig 6, where polyclonal Fabs of the various specificities detected by the 3D analysis are docked onto a reference SOSIP trimer model. Fab epitopes are defined based on the partial overlap with the receptor binding site (CD4bs), variable loops (e.g., V1, V2, V3 and their combinations) and, in some cases, overlap with one or more glycan sites (e.g., N611, N618/N625 and N355/N289). The ConM primary sequence encodes every clade-M consensus PNGS, which implies that there are no large holes in the glycan shield. However, studies of trimers from other genotypes show that some glycan sites are not fully occupied on SOSIP trimers, particularly those on the gp41 subunit [76–78]. There can be substantial antibody responses to glycan holes that are created by under-glycosylation of canonical PNGS. The resolution of negative stain EM does not allow us to conclude whether a glycan is present or not on the ConM-SOSIP trimer, but by extrapolation from studies of trimers of other genotypes, primarily BG505, glycan holes arising from under-glycosylation may be present on the ConM trimers.

For reference, binding and neutralizing antibody titers are also shown for each of the four animals at the two time-points. On a qualitative level, the data indicate that the immunogenic properties of the ConM-SOSIP.v7 trimer were not influenced by how it was presented. This inference is particularly true at week 22 (2 weeks after the final boost) where similar antibody specificities were detected in all four rabbits, notably ones targeting the V1/V2/V3 interface, the N618/N625 and N355/N289 glycan epitopes and the trimer base. The N611 glycan epitope was also targeted in three animals (r2383 was the exception), while CD4bs-associated responses were detected in one animal from each group (r2381 and r2385). The CD4bs is a well-characterized bNAb target that is the focus of several vaccine design strategies [54,79]. We were unable to identify any relationship between the type and quantity of the antibody specificities induced and the binding and NAb responses present in the corresponding sera at the week-22 time point.

There was more variability in the antibody specificities detected at week 4. The N618/N625 and N355/N289 glycan epitopes were targeted in all 4 animals, while anti-base antibodies were induced in both animals from the soluble trimer group and in animal r2383 from the nanoparticle group. Antibodies to the N611 glycan site were detected in one animal from each group. However, antibodies against the V1/V2/V3 interface, located at the top of the trimer, were visible only in the nanoparticle immunogen group at this early time point. The initial antibody response induced after a single immunization (i.e., at week 4) may be skewed towards the V1/V2/V3 epitopes at the trimer apex and away from the trimer base in the nanoparticle immunization group, compared to the soluble trimer group. The early differences in how the antibody responses are primed may contribute to the higher neutralization titers induced in the nanoparticle group after three immunizations (i.e., by week 22).

## Discussion

We evaluated the structure, stability, antigenicity and immunogenicity of two-component nanoparticles presenting HIV Env trimers. In contrast to previous work, the nanoparticles were designed *de novo* to comprise a trimeric building block tailored to support C-terminal

fusion to HIV Env and other trimeric viral antigens [38]. Our library consisted of five nanoparticle candidates of different geometries: three tetrahedral, one octahedral and one icosahedral, presenting 4, 8 and 20 Env trimers, respectively. Two tetrahedral (T33_dn2 and T33_dn10) and one icosahedral nanoparticle (I53_dn5) were able to support the presentation of BG505-SOSIP.v5.2(7S), the model Env trimer selected for optimization. Biophysical and antigenic characterization of the nanoparticle-presented trimers showed that they were folded appropriately, with no detectable adverse effects of nanoparticle incorporation. The nanoparticle-displayed trimers were also glycosylated comparably to their soluble counterparts [18,76]. However, nanoparticle presentation did reduce the accessibility of epitopes on and proximal to the trimer base, which is consistent with antibody binding analyses of other self-assembling nanoparticle designs [37,38].

The tetrahedral nanoparticles (T33_dn2 and T33_dn10) are the lowest valency nanoparticle systems developed for presentation of Env trimer immunogens. Both assembled efficiently and were highly stable under a range of stress-inducing conditions in vitro. They were also superior to soluble trimers when tested in $Ca^{2+}$ flux-based assays of B-cell stimulation. This finding implies that that as few as four BG505-SOSIP antigens are sufficient to meet the minimum threshold for activation when B-cell receptors have high affinity for the antigen (Fig 4). Of note is that the BG505- and ConM-SOSIP-T33_dn2 nanoparticles can be produced at high yield; soluble trimers based on these two genotypes are currently being evaluated for safety and immunogenicity in human clinical trials.

The observed issues with assembly and stability of the icosahedral BG505-SOSIP-I53_dn5 nanoparticles are somewhat unexpected given that these particles assemble very efficiently when Env is not present (assembly completes within ~30 minutes) and when other antigens (i.e., influenza HA and prefusion RSV F) are displayed by genetic fusion [38]. The instability is most likely caused by the crowding of heavily glycosylated Env trimers on the surface of I53_dn5. Additionally, excessive steric clashes could lead to the formation of partially assembled particles that are difficult to distinguish and purify away from the fully assembled ones. It is possible that partially assembled particles constitute the majority of the sample following purification, and that these were unable to withstand the stresses to which they were subjected. Designing nanoparticles with larger diameters and increased spacing to decrease the antigen crowding effect while maintaining the spacing necessary for multivalent interactions with B cells may help achieve more optimal properties with the higher valency particles.

We used rabbits to compare the immunogenicity of the tetrahedral ConM-SOSIP T33_dn2 nanoparticles and the corresponding soluble ConM-SOSIP.v7 trimers (Fig 5 and Fig 6). The anti-trimer binding antibody responses were similar in the two groups, but the autologous NAb titers were higher in the nanoparticle recipients. The results of experiments with heterologous viruses suggest that the elicited neutralizing antibody response is primarily ConM-specific, but it does not appear to be targeting the V3-tip epitope. Polyclonal epitope mapping, performed on 2 animals from each group with the highest neutralizing titers at week-22 time point, suggested that the increased autologous neutralization titers could be due to better shielding of epitopes that are targets for non-neutralizing antibodies (e.g., the trimer base) and/or more efficient priming of antibodies targeting the variable loops located on the top of the ConM trimer (V1/V2/V3). This is supported by the earlier immunogenicity studies performed with ConM-based immunogens that identified variable loops at the ConM apex (V1V2 with some contribution from the V3-base) as the most dominant neutralizing epitopes [8,37]. Furthermore, a meta-analysis of multiple nanoparticle immunization experiments shows that the greatest benefit, compared to soluble trimers, arises when the NAb epitopes are located near the trimer apex [19]. Our data support this conclusion and suggests that immunization platforms based on apex-targeting Env immunogens may benefit from nanoparticle

display. Importantly, however, the immunogenicity data confirm that non-apex epitopes are also accessible and that T33_dn2-presented Env trimers are capable of eliciting antibodies against the same epitopes as free trimers. This provides a rationale for using tetrahedral nanoparticles with a wide range of epitope-focused HIV vaccine design approaches.

Another mechanism may be contributing to the early advantage in the development of neutralizing antibody response seen in the animals that received the nanoparticle immunogen. Recently, we have performed a study using non-human primate animal model to assess the trafficking of T33_dn2 nanoparticle presenting BG505-SOSIP immunogen in comparison to free BG505-SOSIP trimers in immunologically naïve animals [67]. Nanoparticle display of BG505-SOSIP dramatically improved the ability of this immunogen to be retained by the lymph nodes (LN) and to accumulate in the lymph node follicles. This process is dependent on immunogen-induced multimerization of mannose-binding lectin (MBL) and it does not require immune complex (IC) formation [26,67]. Hence, it makes a significant impact on immune response development during the early stages of immunization (i.e. after priming). Significant differences in the neutralizing antibody response observed between the animals receiving ConM-SOSIP-T33_dn2 nanoparticles and ConM-SOSIP.v7 trimers at these early time points may have been a consequence of better trafficking of the nanoparticle immunogen via the MBL pathway. With the development of antigen-specific antibodies at later time points, the IC-mediated pathway may cancel the early trafficking advantage of the nanoparticle immunogen relative to the soluble Env trimer.

Comparison of the immunization data acquired for the tetrahedral T33_dn2 nanoparticle and higher valency ferritin (octahedral) and I53-50 (icosahedral) nanoparticles showed that despite the lower total number of presented trimers, the immunogenicity of the tetrahedral nanoparticle matches the other two systems. As mentioned above, co-display of 4 Env trimers on T33_dn2 is sufficient to trigger the MBL-pathway and increase immunogen accumulation in the LN follicles. This initiates the formation of germinal centers (GCs) within the follicle, where antigens interact with B cells leading to the development of antibody response. Along the same lines, the results of *in vitro* B-cell activation assays described above show that tetrahedral nanoparticles activate antigen-specific B cells comparably to the higher valency particles and superior to free trimers. Altogether, our findings suggest that in addition to being a valuable tool for studying different aspects of the immune response, tetrahedral nanoparticles can help improve the immunogenicity of the presented antigen.

However, nanoparticle presentation of ConM-SOSIP did not provide the expected, strong boost in immunogenicity seen with other viral antigens (influenza HA, DS-Cav1 from RSV) [34,80]. This is something that has already been observed with HIV Env trimer immunogens in different nanoparticle systems [19,61] and the potential causes are discussed in the Introduction section of this paper. In addition to the challenges arising from the densely glycosylated Env immunogens, we believe that the immune-distracting antibody response against the nanoparticle core and potential disassembly of nanoparticles *in vivo* are the main factors limiting the development of stronger immune response against the presented trimer. Additionally, the antibodies against the base of the trimer were still elicited in the nanoparticle group of animals. More studies are required to better understand the exact contributions of these processes that would then drive the optimization of two-component nanoparticle platform.

Overall, our data confirm the ability of designed two-component nanoparticles to at least moderately improve the immunogenicity of HIV Env antigens and provides *in vitro* and *in vivo* evidence of the potential benefits of these systems. Additionally, we introduce and validate two tetrahedral nanoparticle platforms that can be used as immunogens as well as tools for basic structural and immunological experiments. The screening, biophysical and structural

characterization approach that we describe provides a roadmap for generating and evaluating multivalent Env trimer immunogens based on two-component self-assembling scaffolds.

## Materials and methods

### DNA vectors and cloning

Constructs containing BG505-SOSIP.v5.2(7S) or ConM-SOSIP.v7 genes codon-optimized for mammalian cell expression were subcloned into a pPPI4 vector. BamHI and NheI restriction sites were used for insertion of different C-terminal nanoparticle assembly components. Restriction enzymes (BamHI-HF and NheI-HF) and Quick Ligation kit were purchased from New England Biolabs (NEB). Linker insertion and modifications in the antigen-bearing components were achieved using Q5 Site Directed Mutagenesis Kit (NEB). Custom DNA primers produced by Integrated DNA technologies (IDT). Assembly component DNA constructs, codon optimized for bacterial expression were subcloned into a pET28b (+) vector. Protein sequences of all constructs used in this study are shown in S1 Table.

### Expression and purification of BG505-SOSIP, ConM-SOSIP and antigen-bearing components

BG505-SOSIP construct used in this study was engineered to carry a combination of SOSIP. v5.2 (mutations: A501C, T605C, I559P, E64K, A73C, A316W, A561C) [5] and MD39 (mutations: M271I, A319Y, R585H, L568D, V570H, R304V, F519S) [3] stabilizing mutations and glycan knock-ins at positions 241 (mutations: P240T, S241N) and 289 (mutations: F288L, T290E, P291S). For sequence information on BG505-SOSIP construct used in this study see S1 and S2 Tables. ConM-SOSIP.v7 construct had SOSIP.v5.2 and TD8 stabilizing mutations [8,74]. ConM-SOSIP.v9 construct was engineered with SOSIP.v6 [5], MD39 [3] and TD8 [74] stabilizing mutations. For sequence information on ConM-SOSIP constructs see S1 Table and S10B Fig. pPPI4 DNA vectors carrying free SOSIP trimers or SOSIP-based antigen-bearing components were transfected into FreeStyle 293F cells using polyethyleneimine (Polysciences, Inc) as described previously[6]. 6 days post transfection the cells were spun down (7,000 RPM for 1 hour at 4°C) and the supernatant was cleared by vacuum filtration (0.45 μm filtration units, Millipore Sigma). Antigen-bearing components were purified using immuno-affinity column with immobilized PGT145 IgG (Sepharose 4B resin, GE Healthcare Life Sciences). 3 M MgCl$_2$ + 250 mM l-arginine (pH 7.2) buffer was applied for protein elution. Eluate was collected into an equal volume of the SEC buffer (25 mM Tris + 500 mM NaCl + 250 mM l-arginine + 5% glycerol, pH 7.4). Affinity-purified protein was concentrated and buffer exchanged to SEC buffer using Amicon ultrafiltration units, 100 kDa cutoff (Millipore Sigma). Size exclusion chromatography was used as a final purification step (HiLoad 16/600 Superdex S200 pg column). Purified proteins were stored at 4°C.

### Assembly component expression and purification

E coli expression system was used for assembly component production (T33_dn2B, T33_dn10B and I53_dn5A). BL21-DE3 cells (NEB) were transformed with pET28b (+) vector carrying the appropriate gene with a C-terminal His-tag. Following inoculation at 37°C, the cells were incubated in self-inducible media [38] for 18 hours (shaking at 220 RPM, 16°C). Centrifugation (3000 RPM, 30 min, 4°C) was used to harvest the cells. Subsequently the cells were resuspended in TBS buffer (25 mM Tris + 2.7 mM KCl + 137 mM NaCl, pH 7.4; Alfa Aesar / Thermo Fisher Scientific, Cat # J60764) containing cOmplete protease inhibitor cocktail (Sigma Millipore) and lysed using sonication and pressurized cell disruption. Cell lysate was cleared by centrifugation

at 12000 RPM for 1 hour at 4˚C. cOmplete His-Tag Purification Resin (Sigma Millipore) was applied for affinity purification. An additional wash step with 100 ml of detergent-containing buffer (25 mM Tris + 500 mM NaCl + 0.5% N-Dodecyl-β-D-maltoside, pH 7.4) was introduced to remove endotoxin from the samples used for immunogen preparation. Samples were eluted using high imidazole buffer (25 mM Tris + 500 mM NaCl + 500 mM imidazole, pH 7.4). Proteins were then concentrated and buffer-exchanged to SEC buffer (25 mM Tris + 500 mM NaCl + 250 mM l-arginine + 5% glycerol, pH 7.4) using Amicon ultrafiltration units, 10 kDa cutoff (Millipore Sigma). HiLoad 16/600 Superdex S200 pg column was used for final size exclusion purification step.

## Nanoparticle assembly studies

Three assembly reactions containing 5 μg of appropriate antigen-bearing component and equimolar amounts of corresponding assembly component were incubated at different temperatures (4, 25 and 37˚C) for 24 hours. Samples were run on NativePAGE 3–12% BisTris Protein Gels using the dark blue cathode protocol for the NativePAGE Novex Bis-Tris Gel system (Thermo Fisher Scientific). Gels were fixed, de-stained as recommended by the protocol and imaged.

## Differential scanning fluorimetry

Measurements were performed on a Prometheus NT.48 NanoDSF instrument (NanoTemper Technologies) as described previously [81]. Protein and nanoparticle samples were diluted to 0.5 mg/ml in the SEC buffer and loaded into NanoDSF capillaries (in triplicates). $T_m$ measurement range was 20–95˚C at a rate of 1˚C/min. The first derivative curve was calculated from the raw data using the instrument software and the location of the maximum recorded as the $T_m$ value.

## Negative stain electron microscopy

Negative stain electron microscopy experiments were performed as described previously [81,82]. Free nanoparticle components and purified nanoparticles were diluted to 20–50 μg/ml and loaded onto the carbon-coated 400-mesh Cu grid (glow-discharged at 15 mA for 25 s) for 10 s. After the sample was blotted off the grids were negatively stained with 2% (w/v) uranyl-formate for 60 s. Data were collected on a Tecnai Spirit electron microscope, operating at 120 keV. Nominal magnification was set to 52,000 X with a pixel size of 2.05 Å at the specimen plane. Electron dose was adjusted to 25 e-/Å$^2$ and the defocus was set at -1.50 μm. All micrographs were recorded on a Tietz 4k x 4k TemCam-F416 CMOS camera using Leginon automated imaging interface. Initial data processing was performed using the Appion data processing suite. For nanoparticle samples, approximately ~1,000 particles were manually picked from the micrographs and 2D-classified using the Iterative MSA/MRA algorithm. For antigen-bearing components and other trimer samples, 20,000–40,000 particles were auto-picked and 2D-classified using the Iterative MSA/MRA algorithm. For 3D classification and refinement, processing was continued in Relion/3.0 [83]. Maps were segmented and color-coded in UCSF Chimera 1.13 [84].

## Cryo-EM grid preparation

Grids were prepared on a Vitrobot mark IV (Thermo Fisher Scientific). The setting were as follows: Temperature set to 10˚C, Humidity at 100%, Blotting time varied in the 3–7 s range, Blotting force set to 0, Wait time of 10s. BG505-SOSIP-T33_dn10 nanoparticle sample was

concentrated to 1.6 mg/ml and BG505-SOSIP-I53_dn5 nanoparticle was concentrated to 1.7 mg/ml. Lauryl maltose neopentyl glycol (LMNG) at a final concentration of 0.005 mM was used for sample freezing. Quantifoil R 2/1 holey carbon copper grid (Cu 400 mesh) were treated with Ar/O$_2$ plasma (Solarus plasma cleaner, Gatan) for 10s before sample loading. Sample was mixed with the appropriate volume of LMNG solution and 3 µl immediately loaded onto the grid. Following the blot step the grids were plunge-frozen into nitrogen-cooled liquid ethane.

## Cryo-EM data collection

Cryo-grids were loaded into a Talos Arctica TEM (Thermo Fisher Scientific) operating at 200 kV, equipped with the K2 direct electron detector camera (Gatan) and sample autoloader. Total exposure was split into 250 ms frames with a total cumulative dose of ~50 e$^-$/Å$^2$. Exposure magnification of 36,000 was set with the resulting pixel size of 1.15 Å at the specimen plane. For BG505-SOSIP-T33_dn10 nanoparticle imaging the nominal defocus range was -0.6 to -2.0 µm. The range was -0.8 to -2.0 for BG505-SOSIP-I53_dn5. Automated data collection was performed using Leginon software [85]. The data collection details for the acquired datasets are presented in S3 Table.

## Cryo-EM image processing

MotionCor2 [86] was used to align and dose-weight the movie micrographs and the aligned micrographs were uploaded to cryoSPARC 2.9.0 [87]. GCTF was then applied to estimate the CTF parameters. Particles were picked using template picker, extracted and 2D classified. Selected subsets of particles were then transferred to Relion/3.0 [83] for further processing. A reference model was generated using Ab-Initio Reconstruction in cryoSPARC 2.9.0. Multiple rounds of 3D classification and refinement were used to sort out a subpopulation of particles that went into the final 3D reconstructions. Tetrahedral and Icosahedral symmetry restraints were applied for all 3D refinement / classification steps during the processing of BG505-SO-SIP-T33_dn10 and BG505-SOSIP-I53_dn5 datasets, respectively. A soft solvent mask around the nanoparticle core was introduced during the final 3D classification, refinement and post-processing steps in order to eliminate the signal originating from flexibly-linked BG505-SO-SIP.v5.2(7S) trimers. Localized reconstruction v1.2.0 [70] was applied to obtain higher resolution information on the presented antigens. Vectors used for subparticle extraction were defined using sets of Chimera marker coordinates for each geometry (tetrahedral and icosahedral). Part of the signal corresponding to the nanoparticle core was subtracted from aligned particles and the trimer subparticles are extracted. Trimer subparticles are then subjected to 2D and 3D classification using a combination of Relion 3.0 and cryoSPARC 2.9.0 packages. Final subset of clean trimer subparticles was 3D-refined with C3 symmetry. A soft solvent mask around the reconstructed trimer was applied for refinement and post-processing steps. A graphical summary of the data processing approach and relevant statistics are displayed in S5 Fig and S6 Fig. Final post-processed maps, half-maps and masks used for refinement and postprocessing were submitted to The Electron Microscopy Data Bank (EMDB). EMDB IDs: 21183 (I53_dn5 nanoparticle core), 21184 (BG505-SOSIP reconstructed from BG505-SOSI-P_I53_dn5), 21185 (T33_dn10 nanoparticle core), 21186 (BG505-SOSIP reconstructed from BG505-SOSIP_T33_dn10).

## Model building and refinement

B-factor-sharpened maps corresponding to the T33_dn10 nanoparticle core and fused BG505-SOSIP.v5.2(7S) antigen generated in the previous step were used for model building

and refinement. T33_dn10 model from Rosetta design was used for NP core refinement (with tetrahedral symmetry). BG505-SOSIP structure from PDB entry 5CEZ [71] was used as a starting model for trimer refinement (with C3 symmetry imposed). Iterative rounds of Rosetta relaxed refinement [68] and manual refinement in Coot [69] were performed to generate the final structures. EMRinger [88] and MolProbity [89] analysis was applied to evaluate the refined models. The refined models of T33_dn10 nanoparticle core and BG505-SOSIP (reconstructed from BG505-SOSIP-T33_dn10 nanoparticle) were submitted to the Protein Data Bank (PDB). PDB IDs: 6VFK (T33_dn10 nanoparticle core); 6VFL (BG505-SOSIP reconstructed from BG505-SOSIP_T33_dn10).

## Biolayer interferometry

Antibodies (IgG) were diluted in kinetics buffer (DPBS + 0.1% BSA + 0.02% Tween-20) to 5 μg/ml. Trimer and nanoparticle concentrations were normalized based on the molar concentration of the antigen (BG505-SOSIP.v5.2(7S) trimer) in each sample. Final BG505-SOSIP.v5.2 (7S) concentration in the test samples was 500 nM. Free BG505-SOSIP.v5.2(7S) trimer was used as a positive control and a reference. Data were acquired on an Octet Red96 instrument (ForteBio). Antibodies were loaded onto anti-human IgG Fc capture (AHC) biosensors (ForteBio) and moved into the sample solutions at appropriate concentrations. Association and dissociation steps were monitored for 180 s and 300 s, respectively. All data were analyzed using the ForteBio data processing package. Background was corrected by subtracting kinetics buffer dataset (negative control). The resulting binding curves for each antibody were corrected by aligning y-axes to the baseline step immediately preceding the association step and subsequently applying interstep correction between the association and dissociation. Baseline-corrected, aligned binding data were exported to Excel and plotted.

## Surface plasmon resonance

Antigenicity of ConM-SOSIP-T33_dn2 nanoparticles and free ConM-SOSIP.v7 envelope trimer was investigated by using surface plasmon resonance (SPR). All experiments were conducted at 25°C on BIAcore 3000 instrument. HBS-EP (GE healthcare Life sciences) was used as running buffer throughout the analysis. To analyze binding of ConM-SOSIP-T33_dn2 nanoparticles and ConM-SOSIP.v7 trimers in solution, monoclonal antibodies (mAbs) were immobilized on CM3 sensor surface via covalently linked anti-Fc fragment. Affinity-purified goat anti-human IgG Fc (Bethyl Laboratories, Inc.) and goat anti-Rabbit IgG Fc (Abcam, USA) was amine coupled to CM3 surface, to capture human/Macaques and Rabbit mAbs, respectively. Both IgG Fc fragments were captured to a density of 5000 RU, as described elsewhere [37]. ConM-SOSIP-T33_dn2 (5nM), and ConM-SOSIP.v7 trimer (20 nM) were used to analyze their binding to selected mAbs. In each experiment, 3 flow cells were used to capture IgG of mAbs on anti-Fc surface at an average density of 319 RU, with a standard deviation of ± 16 RU (SEM = 1.5 RU), while one flow cell was used as reference. In each cycle, analyte (NP or trimer) was allowed to associate for 300 s, followed by dissociation for 600 s. At the end of each cycle, surface was regenerated with a single pulse of 75 μl of 10 mM Glycine (pH 2.0) at flow rate of 75 μl/min. MW of the peptide portion of ConM-SOSIP.v7 is 325 kDa with glycans and of ConM-SOSIP-T33_dn2 is 1.658 MDa with glycans. Data analysis and interpretation in experiments with nanoparticles and soluble antigens is described elsewhere [37]. In a complementary approach, trimer and NP:s were immobilized by anti-His capture on C1 chips, with background controls and subtractions as described [37,90]. NPs were captured at 374 ± 0.61 RU and trimer at 297 ± 0.23 RU so as to obtain the same amount of Env per surface area.

MAbs were injected at 1uM and allowed to associate for 5min and dissociate for 10min, at flow rate of 30ul/min.

## Nanoparticle stability tests

For pH sensitivity assessment, 3 aliquots of 5 μg of each nanoparticle sample (stored in TBS) were diluted using a set of concentrated Tris/Acetate buffers of different pH (100 mM Tris-Base + 150mM NaCl, pH adjusted with glacial acetic acid to pH 5, 7 and 9) and incubated at room temperature for 1 hour. Salt sensitivity assays were performed by diluting the same amount of each nanoparticle sample into buffers of different NaCl concentration (25 mM Tris + 25 / 150 / 1000 mM NaCl, pH 7.4), and subsequent incubation for 1 hour at room temperature. Temperature sensitivity was probed by incubating nanoparticles in TBS at a range of temperatures (25, 37, 50, 65˚C) for 1 hour. Sensitivity to freeze-thaw was probed by 1 or 2 rounds of flash-freezing in liquid nitrogen for 5 min followed by a gentle thaw at 4˚C for 1 hour. Sample homogeneity was assessed using Native PAGE (NativePAGE 3–12% BisTris Protein Gels), dark blue cathode protocol for NativePAGE Novex Bis-Tris Gel system (Thermo Fisher Scientific). Gels were fixed and de-stained using the recommended protocol and imaged.

## N-glycan analysis using hydrophilic interaction chromatography-ultra-high-performance liquid chromatography (HILIC-UPLC)

N-glycan profiling using HILIC-UPLC has been described in detail elsewhere [76,91]. In short, N-linked glycans were released from gp140 in-gel using PNGase F (New England Bio-labs). The released glycans were subsequently fluorescently labelled with procainamide and excess label and PNGase F was removed using Spe-ed Amide-2 cartridges (Applied Separations). Glycans were analyzed on a Waters Acquity H-Class UPLC instrument with a Glycan BEH Amide column (2.1 mm x 100 mm, 1.7 μM, Waters). Fluorescence was measured, and data were processed using Empower 3 software (Waters, Manchester, UK). The relative abundance of oligomannose glycans was measured by digestion with Endoglycosidase H (Endo H; New England Biolabs). Digested glycans were cleaned using a PVDF protein-binding membrane (Millipore) and analyzed as described above.

## Site-specific glycan analysis using mass spectrometry

Env proteins were denatured for 1h in 50 mM Tris/HCl, pH 8.0 containing 6 M of urea and 5 mM dithiothreitol (DTT). Next, the Env proteins were reduced and alkylated by adding 20 mM iodacetamide (IAA) and incubated for 1h in the dark, followed by a 1h incubation with 20 mM DTT to eliminate residual IAA. The alkylated Env proteins were buffer-exchanged into 50 mM Tris/HCl, pH 8.0 using Vivaspin columns (3 kDa) and digested separately O/N using trypsin or chymotrypsin (Mass Spectrometry Grade, Promega) at a ratio of 1:30 (w/w). The next day, the peptides were dried and extracted using C18 Zip-tip (MerckMilipore). The peptides were dried again, re-suspended in 0.1% formic acid and analyzed by nanoLC-ESI MS with an Easy-nLC 1200 (Thermo Fisher Scientific) system coupled to a Fusion mass spectrometer (Thermo Fisher Scientific) using higher energy collision-induced dissociation (HCD) fragmentation. Peptides were separated using an EasySpray PepMap RSLC C18 column (75 μm x 75 cm). The LC conditions were as follows: 275-minute linear gradient consisting of 0–32% acetonitrile in 0.1% formic acid over 240 minutes followed by 35 minutes of 80% acetonitrile in 0.1% formic acid. The flow rate was set to 200 nL/min. The spray voltage was set to 2.7 kV and the temperature of the heated capillary was set to 40˚C. The ion transfer tube temperature was set to 275˚C. The scan range was 400–1600 m/z. The HCD collision energy was set to 50%, appropriate for fragmentation of glycopeptide ions. Precursor and fragment

detection were performed using an Orbitrap at a resolution $MS_1 = 100,000$. $MS_2 = 30,000$. The AGC target for $MS_1 = 4e^5$ and $MS_2 = 5e^4$ and injection time: $MS_1 = 50$ ms $MS_2 = 54$ ms.

Glycopeptide fragmentation data were extracted from the raw file using Byonic (Version 3.5) and Byologic software (Version 3.5; Protein Metrics Inc.). The glycopeptide fragmentation data were evaluated manually for each glycopeptide; the peptide was scored as true-positive when the correct b and y fragment ions were observed along with oxonium ions corresponding to the glycan identified. The MS data were searched using a standard library for HEK293F expressed BG505 SOSIP.664. The relative amounts of each glycan at each site as well as the unoccupied proportion were determined by comparing the extracted chromatographic areas for different glycotypes with an identical peptide sequence. The precursor mass tolerance was set at 4 ppm and 10 ppm for fragments. A 1% false discovery rate (FDR) was applied. The relative amounts of each glycan at each site as well as the unoccupied proportion were determined by comparing the extracted ion chromatographic areas for different glycopeptides with an identical peptide sequence.

## Site-specific analysis of low abundance N-glycan sites using mass spectrometry

To obtain data for sites that frequently present low intensity glycopeptide the glycans present on the glycopeptides were homogenized to boost the intensity of these peptides. A separate tryptic digest was used for this workflow. This analysis loses fine processing information but enables the ratio of oligomannose: complex: unoccupied to be determined. The peptides were first digested with Endo H (New England Biolabs) to deplete oligomannose- and hybrid-type glycans and leave a single GlcNAc residue at the corresponding site. The reaction mixture was then dried completely and resuspended in a mixture containing 50 mM ammonium bicarbonate and PNGase F (New England Biolabs) using only $H_2{}^{18}O$ (Sigma-Aldrich) throughout. This second reaction cleaves the remaining complex-type glycans but leaves the GlcNAc residues remaining after Endo H cleavage intact. The use of $H_2{}^{18}O$ in this reaction enables complex glycan sites to be differentiated from unoccupied glycan sites as the hydrolysis of the glycosidic bond by PNGaseF leaves a heavy oxygen isotope on the resulting aspartic acid residue. The resultant peptides were purified as outlined above and subjected to reverse-phase (RP) nanoLC-MS. Instead of the extensive N-glycan library used above, two modifications were searched for: +203 Da corresponding to a single GlcNAc, a remnant of an oligomannose/ hybrid glycan, and +3 Da corresponding to the $^{18}O$ deamidation product of a complex glycan. A lower HCD energy of 27% was used as glycan fragmentation was not required. Data analysis was performed as above and the relative amounts of each glycoform determined, including unoccupied peptides

## B cell activation assays

B cell activation experiments were performed as previously described [55]. K46 cells expressing doxycycline-inducible VRC01, PGT145 and PGT121 receptors (in a form of IgM) were grown in advanced Dulbecco's modified Eagle's medium (DMEM) (Gibco), supplemented with 10% fetal calf serum, penicillin/streptomycin antibiotics, and puromycin (2 μg/ml; Gibco). 1 μg/ml Doxycycline was added overnight to induce BCR expression. Cells were spun down (500 g, 3 min, room temperature) and resuspended in RPMI 1640 media supplemented with Gluta-MAX (1X), 10% FBS, penicillin/streptomycin antibiotics (1X), 2-mercaptoethanol, with 1.5 μM Indo-I fluorescent dye (Thermo Fisher Scientific). Cells were incubated for 1 hour with the dye at 37°C, washed 3 times with cold PBS and transferred to fresh cold media that does not contain Indo-I (cell density: $1 * 10^6$ cells/ml). Calcium mobilization was recorder on an

LSR II flow cytometer (BD Biosciences) by measuring the 405/485-nm emission ratio of Indo-1 fluorescence upon UV excitation at room temperature. Trimer and nanoparticle samples were normalized to contain an equimolar amount of BG505-SOSIP.v5.2(7S) antigen in each sample (7.5 nM final assay concentration). Anti-mouse IgM antibody (Jackson ImmunoResearch) at 10 µg/ml was used as a positive control. 1 ml Aliquots of Indo-I treated cells (cell density: $1 * 10^6$ cells/ml) were incubated for 60 s for the baseline signal to be recorded, after which they are stimulated with an antigen for 180 s. Immediately following this step, ionomycin (at 1 µg/ml) is added to the cells and the signal was recorded for another 60 s to verify Indo-I loading. Data analysis was performed using FlowJo (Tree Star, Ashland, OR).

### Ethics statement

All immunization procedures were performed by Covance Research Products Inc. and complied with all relevant ethical regulations and protocols of the Covance Institutional Animal Care and Use Committee under permits with approval number C0171-017 (Denver, PA, USA).

### Immunization experiments

Immunizations and blood draws were performed by Covance Research Products Inc. under permits with approval number C0171-017 (Denver, PA, USA). 2 groups of 5 female New Zealand White Rabbits were immunized at weeks 0, 4 and 20 with ConM-SOSIP.v7 (Group 1, 30 µg/dose) and ConM-SOSIP-T33_dn2 nanoparticle (Group 2, 43 µg/dose) Total dose was normalized to achieve an equivalent molar concentration of ConM-SOSIP.v7 antigen across all animals. Peptidic molecular weight of the protein was used for dose calculations (disregarding glycan). Immunogens were formulated with GLA-LSQ adjuvant (25 µg GLA and 10 µg QS21 per dose; The Infectious Disease Research Institute (IDRI), Seattle, WA). Rabbits were immunized with antigen-adjuvant formulation via intramuscular route. The dose was split in half and injected into both quadriceps. Blood draws were performed at weeks 0, 2, 4, 6, 8, 12, 16, 20 and 22. These experiments were performed in parallel with ConM-SOSIP.v7 immunization experiments reported elsewhere [37] and the data acquired for ConM-SOSIP.v7 control group (Group 1) is shared between two studies.

### ELISA binding assays

Experiments were performed as described previously [37]. ConM-SOSIP.v7 carrying a C-terminal His-tag (diluted to 6.5nM in TBS) was immobilized onto 96-well Ni-NTA ELISA plates (Qiagen) by incubation for 2 hours at room temperature after which the plates were washed 3 times with TBS. Plates were blocked with TBS + 2% skimmed milk and washed 3 times with TBS. For experiments with T33_dn2 nanoparticle core, 6nM final concentration of the antigen was used for plate preparation. Serial three-fold sera dilutions were prepared in the binding buffer (TBS + 2% skimmed milk + 20% sheep serum) starting at a minimum of 1:200. For experiments with T33_dn2 nanoparticle core the starting dilution was 1:1000. Diluted samples were incubated on the plates for 2 hours at room temperature. Following 3 washes with TBS, HRP-conjugated goat anti-rabbit IgG (Jackson Immunoresearch) in TBS + 2% skimmed milk was added for 1 hour at room temperature. Detection antibody was diluted 1:3000. Plates were washed 5 times with TBS + 0.05% Tween-20, followed by the addition of developing solution (1% 3,3',5,5'-tetranethylbenzidine (Sigma-Aldrich) + 0.01% hydrogen peroxide, 100 mM sodium acetate and 100 mM citric acid) was added. Colorimetric endpoint development was allowed to proceed for 3 min before termination by 0.8 M $H_2SO_4$. Endpoint titers were determined using Graphpad Prism software. Titer values are shown in S5 Table.

## Virus neutralization assays

Pseudovirus neutralization assays with Tier-1 Env sequences were performed at Amsterdam University Medical Center (AUMC), The Netherlands, as described previously [92]. The neutralization assays with Tier-2 viruses were performed at Duke University Medical Center in Durham, NC, USA, as described previously [93]. Serial 3-fold dilutions of rabbit sera samples (starting at 1:20 or 1:100 dilution) were prepared and tested against different Env-pseudotyped viruses in TZM-bl cells in a 96-well format. Midpoint titers ($ID_{50}$) were determined using Graphpad Prism software. Autologous $ID_{50}$ titer values are shown in S6 Table, and heterologous $ID_{50}$ titer values are presented in S7 and S8 Tables.

## EM-based polyclonal epitope mapping–Preparation of Fab and complex samples

Experiments were performed as described previously [75]. Briefly, serum samples (weeks 4 and 22) from 2 immunized animals with highest neutralization titers in each group were selected for polyclonal epitope mapping. Animal ID of selected animals: r2381 (Group 1), r2382 (Group 1), r2383 (Group 2) and r2385 (Group 2). IgGs were purified from ~1–3 ml of serum using equal volume of settled Protein A Sepharose resin (GE Healthcare). Samples were eluted off the resin with 0.1 M glycine pH 2.5 and immediately neutralized with 1 M Tris-HCl pH 8. Amicon ultrafiltration units, 10 kDa cutoff (Millipore Sigma) were used to concentrate the purified IgG and buffer exchange to the digestion buffer (PBS + 10 mM EDTA + 20 mM cysteine, pH 7.4). IgG samples were digested for 5 hours at 37˚C using 50 μl of settled papain-agarose resin (Thermo Fisher Scientific). Fc and non-digested IgG were removed by 1-hour incubation with Protein A Sepharose resin, using 0.2ml packed resin per 1 mg of starting IgG amount (room temperature). Fab samples were concentrated to ~2–3 mg/ml using Amicon ultrafiltration units, 10 kDa cutoff (Millipore Sigma), and in the process buffer was exchanged to TBS. Final Fab yields were ~300 μg. Initial assembly trials were performed with 250 μg of purified Fab samples and 15 μg of ConM-SOSIP.v7 for ~18 hours at room temperature, but we observed that Fab binding caused >70% of these trimers to dissociate into gp120-gp41 monomers. Accordingly, it was not possible to use the EMPEM data to reliably assign epitopes (S10A Fig). Instead, we complexed the purified Fabs with ConM-SOSIP.v9 trimers, which contain additional stabilizing mutations that do not affect antigenicity (see Methods, and S1 Table, S10A and S10B Fig). When the more stabilized trimer was used for EMPEM, only <5% of the trimers dissociated into gp120-gp41 monomers after Fab binding, which allowed us to assign the recognized epitopes more reliably (S10A Fig). Hence, the complexing was performed with 250 μg of polyclonal Fab and 15 μg of ConM-SOSIP.v9. Complexes were purified using SEC (Superose 6 Increase column) with TBS as a running buffer, concentrated with Amicon ultrafiltration units (10 kDa cutoff) and immediately loaded onto the carbon-coated 400-mesh Cu grid (glow-discharged at 15 mA for 25 s). Samples were diluted in TBS to 50 μg /ml prior to loading. Grids stained with 2% (w/v) uranyl-formate for 60 s.

## EM-based polyclonal epitope mapping–Sample imaging and data processing

Imaging was performed as described above in the negative stain electron microscopy method section. All initial processing was performed using the Appion data processing package [94]. Approximately, 120,000–150,000 particles are picked and extracted. Particles were then 2D-classified in Relion 3.0 [83] into 250 classes (50 iterations), and particles with complex-like features (~70–90% of the starting number) were selected for 3D sorting in Relion 3.0. A low-

resolution model of non-liganded HIV Env ectodomain was used as a reference for all 3D steps. Initial 3D classification was performed with 40 classes. Particles from similar looking classes were then pooled and reclassified. A subset of 3D classes with unique structural features (in terms of Fab specificities) was subjected to 3D auto-refinement in Relion 3.0. Maps from 3D refinement were loaded into UCSF Chimera 1.13 [84] for visualization, segmentation and figure preparation. 3D refinement was performed on a subset of 2D-cleaned particles (following the initial 2D classification step and before any 3D classification) and the refined model is submitted to EMDB. The list of EMDB IDs: 21175 (ConM-SOSIP.v9 + Wk4-r2381 polyclonal Fab); 21176 (ConM-SOSIP.v9 + Wk4-r2382 polyclonal Fab); 21177 (ConM-SOSIP.v9 + Wk4-r2383 polyclonal Fab); 21178 (ConM-SOSIP.v9 + Wk4-r2385 polyclonal Fab); 21179 (ConM-SOSIP.v9 + Wk22-r2381 polyclonal Fab); 21180 (ConM-SOSIP.v9 + Wk22-r2382 polyclonal Fab); 21181 (ConM-SOSIP.v9 + Wk22-r2383 polyclonal Fab), 21182 (ConM-SOSIP.v9 + Wk22-r2385 polyclonal Fab). Full particle stacks and 3D models used for Fab segmentation and generation of composite figures are available upon request.

## Supporting information

**S1 Table. Protein sequences of constructs used in this study.**
(DOCX)

**S2 Table. The list of stabilizing and glycan knock-in mutations in different versions of BG505 SOSIP (MD39, v5.2 and v5.2(7S)).**
(DOCX)

**S3 Table. Cryo EM data collection statistics.**
(DOCX)

**S4 Table. Model building and refinement statistics for T33_dn10 nanoparticle core and presented BG505-SOSIP.**
(DOCX)

**S5 Table. Anti-trimer binding antibody titers against ConM-SOSIP.v7.**
(DOCX)

**S6 Table. Autologous neutralization titers ($ID_{50}$) against ConM-based pseudovirus.** Midpoint neutralization titers ($ID_{50}$) were determined using TZM-bl neutralization assays and sera samples from different time points (indicated in the top row). Neutralization assays with murine leukemia virus (MLV) were also performed (negative control). Color coding: white = no neutralization ($ID_{50} < 20$); yellow = very weak neutralization ($20 < ID_{50} < 100$); light orange = moderate neutralization ($100 < ID_{50} < 1000$); dark orange = strong neutralization ($1000 < ID_{50} < 10000$); red = very strong neutralization ($ID_{50} > 10000$).
(DOCX)

**S7 Table. Neutralization titers ($ID_{50}$) of week-22 sera samples against the selected heterologous Env-pseudotyped viruses.** The name and tier classification for each HIV Env sequence is indicated. Color coding: white = no neutralization ($ID_{50} < 20$); yellow = very weak neutralization ($20 < ID_{50} < 100$); light orange = moderate neutralization ($100 < ID_{50} < 1000$); dark orange = strong neutralization ($1000 < ID_{50} < 10000$); red = very strong neutralization ($ID_{50} > 10000$). Toxicity was observed at 1:20 dilution for all samples highlighted in gray.
(DOCX)

**S8 Table. Heterologous neutralization titers ($ID_{50}$) against viruses pseudotyped with Tier 1 HIV Env sequence.** Color coding: white = no neutralization ($ID_{50} < 20$); yellow = very weak

neutralization ($20 < ID_{50} < 100$); light orange = moderate neutralization ($100 < ID_{50} < 1000$); dark orange = strong neutralization ($1000 < ID_{50} < 10000$); red = very strong neutralization ($ID_{50} > 10000$).
(DOCX)

**S1 Fig. Nanoparticle library evaluated in this study.** (a) Structural models of nanoparticle candidates derived from Rosetta_design. For clarity, trimeric antigen-bearing component is shown in orange and assembly component in blue. (b) Geometric properties of different nanoparticle candidates. (c) Nanoparticle naming system explained on the example of I53_dn5.
(TIF)

**S2 Fig. Purification and characterization of different antigen-presenting components and assembled nanoparticles.** (a) SEC curves of BG505-SOSIP.v5.2(7S) and BG505-SOSIP-fused nanoparticle components. (b) SDS PAGE analysis of the purified assembly component for T33_dn2, T33_dn10 and I53_dn5 nanoparticle systems. (c) Extended data on BLI characterization of the antigenicity of the three antigen-bearing components compared to BG505-SOSIP.v5.2(7S) using 19b and F105 antibodies. (d) SEC purification of different nanoparticle candidates after assembly. (e) SDS PAGE gel of the purified nanoparticles confirming the presence of both, antigen-bearing and assembly components. (f) Extended data on BLI characterization of the antigenicity of the three nanoparticle systems compared to BG505-SOSIP.v5.2 (7S) using 19b and F105 antibodies.
(TIF)

**S3 Fig. Site specific glycan analysis of BG505-SOSIP-bearing components and free BG505-SOSIPv5.2(7S).** The table shows the glycoforms found at each potential N-linked glycosylation site (PNGS), compositions corresponding to oligomannose/hybrid-type glycans are colored green and fully processed complex type glycans are colored magenta. PNGS with no attached glycan are colored grey. Oligomannose-type glycans are categorized according to the number of mannose residues present, hybrids are categorized according to the presence/absence of fucose and complex-type glycans are categorized according to the number of processed antenna and the presence/absence of fucose. Sites that could only be obtained from low intensity peptides cannot be distinguished into the categories in the table and so are merged to cover all oligomannose/hybrid compositions or complex-type glycans.
(TIF)

**S4 Fig. Nanoparticle stability studies.** Native PAGE assays were used for evaluation of nanoparticle integrity following the incubation under the specified conditions.
(TIF)

**S5 Fig. Cryo-EM analysis of BG505-SOSIP-T33_dn10 nanoparticle.** Schematic representation of the data processing workflow with relevant statistics.
(TIF)

**S6 Fig. Cryo-EM analysis of BG505-SOSIP-I53_dn5 nanoparticle.** Schematic representation of the data processing workflow with relevant statistics.
(TIF)

**S7 Fig. ConM-SOSIP-T33_dn2 nanoparticle purification and characterization.** (a) SEC purification of ConM-SOSIP-T33_dn2A and 2D class-averages from negative-stain-EM analysis. (b) SEC purification of assembled ConM-SOSIP-T33_dn2 and NS-EM analysis of the purified nanoparticles (raw micrograph and 2D class averages). (c) SPR-based characterization of the antigenicity of purified nanoparticles with immobilization of monoclonal antibodies

(antigens were in the mobile phase). ConM-SOSIP.v7 trimer was used as a reference. In addition to affinity, SPR signal is also a function of antigen size (molecular weight). MW of the ConM-SOSIP-T33_dn2 nanoparticle is ~5.1 times higher than that of soluble ConM-SOSIP.v7 trimer. See methods section for data analysis information. (d) SPR-based characterization of the antigenicity of purified nanoparticles with immobilization of antigens (antibodies were in the mobile phase). ConM-SOSIP.v7 was used as a reference.
(TIF)

**S8 Fig. Extended immunization data.** (a) Anti-nanoparticle core response (ELISA binding titers) in individual Group 2 animals, with the mean value indicated by the solid line. The dashed line represents the assay detection limit. (b) Ratios of NAb titers and anti-trimer binding antibody titers in sera from individual animals in Group 1 (open circles) and Group 2 (black squares) were calculated at weeks 4, 8 and 22. Scatter plots are shown with mean values indicated. Red asterisks indicate sera samples where neutralization titers were below the level of detection (1:20 titer). An AUC statistical analysis of the titer ratio values for Group 1 versus Group 2 as a function of time results in p = 0.016.
(TIF)

**S9 Fig. Comparison of the immunization data for tetrahedral (T33_dn2), octahedral (ferritin) and icosahedral (I53-50) nanoparticle platforms presenting ConM-SOSIP.v7.** Neutralizing antibody titers ($ID_{50}$) in each animal at different time points are presented as scatter plots with the mean titers indicated by lines. Data corresponding to ConM-SOSIP.v7 (open circles), ConM-SOSIP-ferritin (black triangles) and ConM-SOSIP-I53-50 (black circles) immunogens was adapted from Brouwer et al., 2019 [37]. Data depicted with black squares corresponds to ConM-SOSIP-T33_dn2 nanoparticle immunogen. (a) Autologous neutralization titers ($ID_{50}$) against the pseudotyped ConM Env virus (sera samples from weeks 4, 8 and 22). (b) Neutralization titers ($ID_{50}$) against the selected heterologous Tier-1 viruses (sera samples from week 22 time point).
(TIF)

**S10 Fig. Comparison of ConM-SOSIP.v7 and ConM-SOSIP.v9.** (a) EMPEM data derived using ConM-SOSIP.v7 (top) and ConM-SOSIP.v9 (bottom), complexed with polyclonal Fab sample isolated from the serum of rabbit r2381 (Grp1) at week 22 time point. Representative raw micrographs (left), 2D classes (middle) and 3D classes (right) are shown. Red circles mark the 2D classes of ConM-SOSIP.v7 gp120-gp41 monomers bound to one or more Fabs. Reconstructed monomer-like 3D classes are shown. Fabs are easily discernable in most 3D classes but epitope assignment is very challenging due to high degree of heterogeneity. Complexing with ConM-SOSIP.v9 results in significantly lower percentage of disassembled trimers (i.e. monomers), which can be observed in 2D and 3D classes. (b) Comparison of stabilizing mutations in different ConM-SOSIP constructs.
(TIF)

**S11 Fig. Extended data for EMPEM analysis of antibody responses in immunized rabbits.** Sample micrograph, 2D class averages and sample 3D reconstructions obtained for antibodies isolated from the specified rabbits at (a) week 4 and (b) week 22.
(TIF)

## Acknowledgments

The authors thank Bill Anderson, Hannah L Turner and Charles A Bowman for their help with electron microscopy, data acquisition and processing. We are also grateful to Kimmo

Rantalainen for providing a small-scale screening protocol for expression and evaluation of fusion constructs. We want to thank Nathan Newton and Zack Korzen-Varin for preforming the neutralization assays with heterologous Env sequences. Finally, we want to acknowledge Lauren Holden for her help with the preparation of this manuscript.

## Author Contributions

**Conceptualization:** Aleksandar Antanasijevic, Thomas J. Ketas, Max Crispin, David Nemazee, John P. Moore, Rogier W. Sanders, Neil P. King, David Baker, Andrew B. Ward.

**Data curation:** George Ueda, Philip J. M. Brouwer, Jeffrey Copps, Deli Huang, Joel D. Allen, Christopher A. Cottrell, Anila Yasmeen, Leigh M. Sewall, Ilja Bontjer, Hannah L. Turner, Zachary T. Berndsen, Max Crispin, David Nemazee, Rogier W. Sanders, Neil P. King, Andrew B. Ward.

**Formal analysis:** Aleksandar Antanasijevic, Deli Huang, Anila Yasmeen, Leigh M. Sewall, Ilja Bontjer, Hannah L. Turner, David C. Montefiori, Per Johan Klasse.

**Funding acquisition:** David C. Montefiori, Per Johan Klasse, Max Crispin, David Nemazee, John P. Moore, Rogier W. Sanders, Neil P. King, David Baker, Andrew B. Ward.

**Investigation:** Aleksandar Antanasijevic, George Ueda, Jeffrey Copps, Deli Huang, Joel D. Allen, Anila Yasmeen, Leigh M. Sewall, Ilja Bontjer, Hannah L. Turner, Zachary T. Berndsen, Per Johan Klasse.

**Methodology:** Aleksandar Antanasijevic, George Ueda, Jeffrey Copps, Deli Huang, Joel D. Allen, Anila Yasmeen, Leigh M. Sewall, Ilja Bontjer, Hannah L. Turner, Per Johan Klasse.

**Project administration:** Aleksandar Antanasijevic, Andrew B. Ward.

**Resources:** David C. Montefiori, Per Johan Klasse, Max Crispin, David Nemazee, Rogier W. Sanders, Neil P. King, Andrew B. Ward.

**Supervision:** David C. Montefiori, Per Johan Klasse, David Nemazee, John P. Moore, Rogier W. Sanders, David Baker, Andrew B. Ward.

**Validation:** Aleksandar Antanasijevic, Andrew B. Ward.

**Visualization:** Aleksandar Antanasijevic, Joel D. Allen.

**Writing – original draft:** Aleksandar Antanasijevic, Andrew B. Ward.

**Writing – review & editing:** George Ueda, Philip J. M. Brouwer, Jeffrey Copps, Deli Huang, Joel D. Allen, Christopher A. Cottrell, Leigh M. Sewall, Ilja Bontjer, Hannah L. Turner, Zachary T. Berndsen, Per Johan Klasse, Max Crispin, David Nemazee, John P. Moore, Rogier W. Sanders, Neil P. King, David Baker, Andrew B. Ward.

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
