## [Decision Letter · Decision Letter 0]

6 Apr 2020

Dear Dr. Ward,

Thank you very much for submitting your manuscript "Structural and functional evaluation of de novo-designed, two-component nanoparticle carriers for HIV Env trimer immunogens" for consideration at PLOS Pathogens. As with all papers reviewed by the journal, your manuscript was reviewed by members of the editorial board and by several independent reviewers. In light of the reviews (below this email), we would like to invite the resubmission of a significantly-revised version that takes into account the reviewers' comments.

Dear Dr. Ward,

I am sorry for the prolonged review. Not surprisingly, we had some delays as a result of the disruption COVID-19 is causing. Although the reviewers generally felt that technical aspects of the study are sound and the manuscript is well-written, they were disappointed that presentation of the SOSIP trimers as nanoparticle assemblies was not more immunogenic. While reviewer 1 was more favorable, reviewers 2 and 3 are not convinced that the study represents a significant advance. Having read the paper myself, I understand the criticisms that were raised. However, as a technical tour de force that illustrates not only the advantages, but also some of the limitations, of nanoparticle Env trimer assemblies as a vaccine approach, I think these results are important and of sufficient interest for PLoS Pathogens. I therefore ask that you carefully address each of the reviewers' comments and revise your manuscript accordingly.

Best regards,

David

We cannot make any decision about publication until we have seen the revised manuscript and your response to the reviewers' comments. Your revised manuscript is also likely to be sent to reviewers for further evaluation.

Sincerely,

David T. Evans

Associate Editor

PLOS Pathogens

Thomas Hope

Section Editor

PLOS Pathogens

Kasturi Haldar

Editor-in-Chief

PLOS Pathogens

orcid.org/0000-0001-5065-158X

Michael Malim

Editor-in-Chief

PLOS Pathogens

orcid.org/0000-0002-7699-2064

Dear Dr. Ward,

I am sorry for the prolonged review. Not surprisingly, we had some delays as a result of the disruption COVID-19 is causing. Although the reviewers generally felt that technical aspects of the study are sound and the manuscript is well-written, they were disappointed that presentation of the SOSIP trimers as nanoparticle assemblies was not more immunogenic. While reviewer 1 was more favorable, reviewers 2 and 3 are not convinced that the study represents a significant advance. Having read the paper myself, I understand the criticisms that were raised. However, as a technical tour de force that illustrates not only the advantages, but also some of the limitations, of nanoparticle Env trimer assemblies as a vaccine approach, I think these results are important and of sufficient interest for PLoS Pathogens. I therefore ask that you carefully address each of the reviewers' comments and revise your manuscript accordingly.

Best regards,

David

Reviewer's Responses to Questions

**Part I - Summary**

Reviewer #1: This well-written study seeks to address some of the drawbacks of soluble HIV Env trimer vaccine platforms – to generate nanoparticles that might enhance immunogenicity, e.g. by providing a multivalent array of trimers, helper epitopes, modified trafficking and/or blocking the hyperimmunogenic foot of these trimers. Composite immunogens were assembled from two components: SOSIP fused to an antigen-bearing component coupled with cognate, computationally designed assembly components. 5 initial formats were reduced to three that properly assemble for antigenic and structural analyses. Interestingly, the nanoparticle constucts showed an ability to stimulate nAb precursor expressing cell lines. One construct was then tested for immunogenicity in rabbits, where similar NAb titers were (slightly elevated) elicited as compared to soluble monomers. Overall, the work is high in concept, but a payoff yet remains unclear. One could compare this to a Prius (SOSIP) remodeled to swap the body with that of a race car (nanoparticle), which ultimately drives the same as the original Prius (immunogenicity) - though the reality may be somewhere in between. A few specific comments follow:

It should be explained why the authors used a tier 1 ConM virus for immunogenicity? This might induce V3 responses that don’t inform vaccine design for efforts to develop products that might be broiadly effective against diverse circulating strains. In the mapping studies, it would be useful to do V3 peptide interference to determine how much this contributes to neutralization titers. It would be better to use BG505 SOSIP – a tier 2 trimer. Related to this, fig S7c, where are V3 mAbs binding data? Also, what is MAb 8D9?? 35022 binding seems not to be reduced against particles.

On the serum mapping, the data implies glycan epitopes like N611, N625 etc, but are glycan contacts actually made or are these glycans just use to assign regions and the sera actually bind underlying protein? Since glycan holes are typically preferred, are there any in the ConM sequence and to what extent do NAbs target these holes? (i.e. check by hole filling, etc).

line 29: “fit the requirements of different immunization strategies” this implies that something is known about what immunization strategies might succeed. Is this known? Hence, it might be better to say to “test novel multimerization concepts in immunization strategies to potentially improve antibody precursor stimulation and neutralizing responses”.

In rabbits, the differences in NAb titers/specificities did not correlate well with the promise of the antigenic stimulation studies seen using cell lines. This may be because the epitopes targeted by rabbits are type specific (V3?) and differ from these prototype antibodies that may be harder to induce. It would be useful to cover this point.

Checking the immunogenenicity of separate SOSIP+unassembled scaffolds together in rabbits would have been a better comparator than just isolated SOSIP (fig 5) to determine if the modest benefit in NAb titers might come from providing helper epitopes in co-immunization. This control would have been especially useful, considering the anti-base antibodies recovered in the nanoparticle group, suggesting disassembly, raising questions about whether the modest NAb differences may be related simply to the co-presence of the two components in the immunizations (rather than their assembled particle). For example, helper epitopes in the scaffolds co-immunized could account for the differences.

The review of nanoparticle approaches and the concept to provide co-stimulation signals is good. However, what is not clear is what the mentioned “appropriate” antigenic spacing actually means and how these constructs meet those needs. The particle size, antigen spacing, flexibility, possible lateral movement of trimers (as possible in membranes) could be important to ensure crosslinking of more than one surface IgM on a precursor cell. Calcium flux experiments seemed to be supportive, but it is unclear if these had much bearing in immunogenicity experiments (as covered above).

Related to the above, Line 104: re multivalent geometry and spacing “This may be particularly important for vaccine design efforts focusing on a specific set of HIV Env epitopes, which need to be presented in an appropriate and accessible orientation.” I am not sure what this sentence implies. What scenarios come to mind? What specific set of epitopes and how might the strategy be tailored?

Line 40: The statement “increased the proportion of the overall antibody response directed against autologous neutralizing Ab epitopes present on the ConM-SOSIP trimers” is a bit ambiguous. A better concluding sentence in the introduction could be something like “Although neutralizing antibody responses exhibited a similar (slightly elevated) frequency titer and breadth (with a greater initial apex focus)”.

Line 53: How can immunogenicity in vitro = precursor cell stimulation? In vitro immunogenicity seems a contradictory term and could be reworded. There should be something straightforward on whether NAb responses were affected, as in the abstract.

Line 94: “precisely defined”… seems less jarring/repetitive sub with “well-defined”

The authors could briefly mention more about the success or not of various nanoparticle approaches in the paragraph at line 107. So far, have they been any better or not? Equivalent? Promising? Variable? Where are the knowledge gaps? The statement that they “provide additional (vaccine/antigen) options” (line 111) is a bit vague, given that there are published studies that should by now give some idea whether these options are actually useful.

S1 Fig/Table 1 Clarity question: why are assembly components in blue sometimes component A for tetrahedral scaffolds and other times component B? I don’t see why they should be switched? I am not sure if the nomenclature of the various antigen bearing and assembly components needs to be explained somehow better. As it stands it is confusing so an explanation of what A and B means and the other parts of name would help maybe as a brief nomenclature or label the components. I don’t know what the 33, 43 the dn means in T33 O43_dn18

Although the v5.2 mutations and MD39 ones may be published, I would suggest they be listed here or highlighted in the sequence in the supp as it should take little space and will save the less committed but curious reader from needing to access other papers and makes this paper more self-contained in the process. It is curious that 14e is not exposed here (Fig 1) whereas it is highly exposed in other preps of SOSIP (e.g. Sanders JBC paper 2019). Whether this is due to an aromatic residue masking the V3 tip as reported previously or a genuine occlusion of the V3 is not clear on the face of the information accessible in this paper. As mentioned above, the V3 exposure on the ConM SOSIP is missing.

Line 201: not dn2A, but dn5B in Fig S3 (all mannose)

Fig 4 colors are not very distinct, too dull to discern types

The comprehensive listing of their SOSIP human clinical trials in the introduction is not necessary to repeat in discussion.

Reviewer #2: In this manuscript by Antanasijevic et al., the authors characterize multimeric nanoparticle immunogens derived from BG505 and ConM SOSIPs, using a two-component, self-assembling, nanoparticle platform. The authors designed and characterized a nanoparticle library consisting of five different constructs based on BG505 SOSIP.v5.2(7S): three tetrahedral (T33_dn2, T33_dn5, T33_dn10), one octahedral (O43_dn18), and one icosahedral (I53_dn5). Of the five constructs, O43_dn18 did not express and T33_dn5 did not assemble efficiently. The three remaining antigen-bearing components were characterized for antigenicity through antibody binding, negative stain electron microscopy (only T33_dn10 and I53_dn5), and glycan profiling and were found to be similar to the parental BG505 SOSIP in key parameters. The T33_dn2 construct was somewhat surprisingly recognized to similar levels by base binding antibodies as the free SOSIP. When the antigen-bearing components were assembled, I53_dn5 did not assemble efficiently. However, all of the constructs had better activation of B cells expressing PGT145, VRC01, and PGT121 BCRs compared to wildtype, although T33_dn2 was the least efficient in all three cases. Nevertheless, the T33_dn2 construct was selected as the platform to display a different SOSIP, CON-M, for immunogenicity testing in rabbits. The free CON-M SOSIP and T33_dn2 immunogens elicited similar serum binding antibody and neutralizing antibody levels, and antibody specificities detected by negative stain EM mapping were also similar in the subset of animals analyzed, including the presence of base binding antibodies.

Overall, the paper is technical in nature and the majority of the paper is devoted to describing the generation and characterization of a panel of multimeric SOSIP antigens. The rationales for moving forward with the T33_dn2 platform and for switching from BG505 to the less characterized/optimized CON-M multimer were not clear. Since the T33_dn2 CON-M immunogen produced binding and neutralizing antibody titers that were very similar to the free trimer, there is no significant advance in immunogen design provided nor is there a compelling reason presented to pursue this approach that would be of interest to a broad readership.

Reviewer #3: Antanasijevic, Ueda et al brings together a number of tools – recombinant, native-like HIV-1 Env trimers and two-component, self-assembling nanoparticles – to generate multivalent vaccine candidates. The general design and structural characterization of these kinds of nanoparticles are described in a bioRxiv paper by the same group (Ref. 38, Ueda, Antanasijevic et al). This manuscript focuses on detailed characterization of the class of particles that display HIV-1 Env spikes. Initial studies were done on the BG505-SOSIP trimer, and then the authors switched to consensus group M (ConM)-SOSIP trimers. Expression and assembly data on various test constructs, BLI binding experiments to test antigenicity, and glycan composition analyses are shown. CyoEM structures of tetrahedral and icosahedral particle types were determined, which confirm the expected structures of both the nanoparticle module and trimer modules. The capacity of the nanoparticles to stimulate antigen-specific B cells was evaluated; it was shown that assembled nanoparticles activated the B cells more strongly, compared to the control (free trimers). Finally, immunogenicity of the ConM-SOSIP nanoparticles was determined in rabbits.

In general, the overall study seems well-designed and executed, and the manuscript presents it in a logical and comprehensible manner. In vitro antigenicity profiles were only done in a qualitative manner (no affinities were determined by BLI), yet the qualitative data are sufficient to show that the nanoparticle design left much of the spike available for presentation and identify occluded regions. The structural analysis seems solid. If the goal of the authors is to present these nanoparticles as viable vaccine candidates then they have succeeded; in this regard the impact and interest of the manuscript is primarily as a proof-of-principle study that may end up occupying a prominent position among the many similar such studies available in the literature. In this reviewer’s opinion, it is primarily a technical paper (with the most interesting aspects already described in the bioRXiv paper, the final form of which will be presumably published elsewhere); thus it falls somewhat short of the kinds of mechanistic and biological insights that one might expect from a paper in PLoS Pathogens.

**Part II – Major Issues: Key Experiments Required for Acceptance**

Reviewer #1: (No Response)

Reviewer #2: 1. The authors should provide a stronger rationale of why BG505 SOSIP was used for structural and in vitro functional characterization and optimization studies but ConM SOSIP was used in the in vivo studies.

2. The authors characterized the antigen-bearing component using a panel of Nabs and two non-Nabs, RM19R and RM20A3. These non-Nabs were included because they are base binding antibodies that are not expected to bind to the antigen-bearing component because this is where the antigen fuses to the nanoparticle. Two of the antigen-bearing components did not bind these antibodies but T33_dn2A showed binding similar to free BG505-SOSIP.v5.2(7S). The authors need to elaborate on why this occurred and support their selection of this construct to move forward.

3. The authors fail to address why cryo-EM was not performed on BG505-SOSIP-T33_dn2A. This is especially important because this platform was chosen to move forward in the rabbit studies.

4. The authors point out a difference in early targeting of the V2 apex in the T33_dn2 immunized rabbits. However, only 2 animals from each group were analyzed by NSEM, and all animals had these antibodies by wk 22, so this conclusion is not supported by the data presented.

5. In Fig. S3, the text does not match the numbers shown for some of the glycan differences (ex. N637, N355), line 199. It appears that dn2A and dn5B could have been inadvertently switched.

6. The 8D9 and 3BC315 antibodies bound less efficiently to the CON-M SOSIP T33_dn2 than to the free trimer but 35022 bound better, (Fig. S7) which is not what is stated in the text on line 336.

7. There is no negative control included in the neutralization assays. Given such high autologous titers, it seems that a negative control virus and purification of IgG would be prudent.

8. Given the use of CON-M SOSIP, was neutralization breadth measured? Presumably that is a reason for switching to CON-M.

9. Given the desire to elicit differences in recognition of neutralizing antibody epitopes, envelope mutants should be used to determine whether there were in fact any distinctions between the free and T33_dn2 immunogens. This could shed light on whether the V2 antibodies seen in the NSEM analysis made any contributions to neutralization.

Reviewer #3: (No Response)

**Part III – Minor Issues: Editorial and Data Presentation Modifications**

Reviewer #1: (No Response)

Reviewer #2: (No Response)

Reviewer #3: (No Response)

PLOS authors have the option to publish the peer review history of their article (what does this mean?). If published, this will include your full peer review and any attached files.

Reviewer #1: No

Reviewer #2: No

Reviewer #3: No
---

## [Editor Report · Decision Letter 1]

28 May 2020

Dear Dr. Ward,

We are pleased to inform you that your manuscript 'Structural and functional evaluation of de novo-designed, two-component nanoparticle carriers for HIV Env trimer immunogens' has been provisionally accepted for publication in PLOS Pathogens.

Best regards,

David T. Evans

Associate Editor

PLOS Pathogens

Thomas Hope

Section Editor

PLOS Pathogens

Kasturi Haldar

Editor-in-Chief

PLOS Pathogens

orcid.org/0000-0001-5065-158X

Michael Malim

Editor-in-Chief

PLOS Pathogens

orcid.org/0000-0002-7699-2064
---

## [Editor Report · Acceptance letter]

23 Jul 2020

Dear Dr. Ward,

We are delighted to inform you that your manuscript, "Structural and functional evaluation of de novo-designed, two-component nanoparticle carriers for HIV Env trimer immunogens," has been formally accepted for publication in PLOS Pathogens.

Best regards,

Kasturi Haldar

Editor-in-Chief

PLOS Pathogens

orcid.org/0000-0001-5065-158X

Michael Malim

Editor-in-Chief

PLOS Pathogens

orcid.org/0000-0002-7699-2064